# Birds have peramorphic skulls, too: anatomical network analyses reveal oppositional heterochronies in avian skull evolution

Olivia Plateau[1] & Christian Foth [1✉]

In contrast to the vast majority of reptiles, the skulls of adult crown birds are characterized by a high degree of integration due to bone fusion, e.g., an ontogenetic event generating a net reduction in the number of bones. To understand this process in an evolutionary context, we investigate postnatal ontogenetic changes in the skulls of crown bird and non-avian theropods using anatomical network analysis (*AnNA*). Due to the greater number of bones and bone contacts, early juvenile crown birds have less integrated skulls, resembling their non-avian theropod ancestors, including *Archaeopteryx lithographica* and *Ichthyornis dispars*. Phylogenetic comparisons indicate that skull bone fusion and the resulting modular integration represent a peramorphosis (developmental exaggeration of the ancestral adult trait) that evolved late during avialan evolution, at the origin of crown-birds. Succeeding the general paedomorphic shape trend, the occurrence of an additional peramorphosis reflects the mosaic complexity of the avian skull evolution.

[1] Department of Geosciences, University of Fribourg, Chemin du Musée 6, CH-1700 Fribourg, Switzerland. ✉email: christian.foth@gmx.net

Birds represent highly modified reptiles and are the only surviving branch of theropod dinosaurs. In contrast to their non-avian theropod ancestors, which possess a typical diapsid skull morphology[1], adult crown birds have highly apomorphic skulls, characterized by a toothless beak, enlarged round orbits, an enlarged and highly pneumatized chrondrocranium, the loss and fusion of bones and skull openings, and a complex kinetic system that allows the simultaneous motion of both jaws, which magnitude is, however, restricted by the morphology and articulation of the quadrate and palate[2,3]. In contrast, Mesozoic Avialae outside the crown, such as the Late Jurassic *Archaeopteryx lithographica* or the Late Cretaceous *Ichthyornis dispars*, still retain numerous ancestral theropod characters[4–6]. A recent comparison of ontogenetic series of non-avian theropods and extant crown birds using geometric morphometrics demonstrated that avian skull shape is the result of a sequence of at least four paedomorphic events in the evolution of Eumaniraptora, meaning that the shape of adult bird skulls retain juvenile features like the enlarged orbit and associated brain regions[7,8]. Another recent application of geometric morphometrics showed that the avian cranium has extensive variational modularity, consisting of seven to eight semi-independent regions evolving in a mosaic pattern[9]. However, when compared to their non-avian theropod ancestors, the number of modules is significantly reduced[10]. Although less drastic, a similar relationship was found for the connectivity modularity in the skull of the non-avian theropod *Tyrannosaurus rex* and crown bird *Gallus gallus*, which based on anatomical network analysis (AnNA)[11] (Fig. 1a, b).

The cranial and mandibular bone fusion that characterizes the skull of adult crown birds mainly occurs during postnatal development, when existing sutures between neighbouring bones are fully closed[12,13]. Consequently, the skull bone configuration of extant bird hatchlings resembles, to a certain degree, that of non-avian theropods, implying a less integrated skull network with high connectivity modularity (Fig. 1c). To test this hypothesis, we investigated the contact and fusion patterns of skull bones and their impact on modularity during ontogeny in 41 extant birds and compared them with those of non-avian archosaurs (including 15 adult and two juvenile non-avian dinosaurs and an ontogenetic pair of *Alligator mississippiensis*; see Supplementary Data 1 and 2 file), using AnNA and phylogenetic comparative analyses (see the "Methods" section).

Our analyses show that early juvenile crown birds have less integrated skulls than adult birds in terms of connectivity due to the greater number of bones and bone contacts, but integration increases continuously with the net reduction of bones during maturation due to fusion. The skulls of early juvenile crown birds, however, resemble those of non-avian theropods, including *Archaeopteryx lithographica* and *Ichthyornis dispars*. In this context, phylogenetic comparisons indicate that the highly integrated adult bird skull evolved late during avian evolution, at the origin of crown-birds, and are a result of a peramorphosis (developmental exaggeration of the ancestral adult trait), which might be related to the origin of cranial kinesis. The sequential occurrence of oppositional heterochronies (i.e., a trended skull shape paedomorphosis within Coelurosauria followed by peramorphic skull bone fusion in the last common ancestor of crown birds) reflects the mosaic complexity of avian skull evolution, facilitating shape and ecological diversity.

## Results

### Anatomical network analysis

The skulls of adult non-avian archosaurs, including *Archaeopteryx* and *Ichthyornis*, differ significantly from adult crown birds in the number of bones ($N$) and bone contacts ($K$), density of connections ($D$), mean shortest path length ($L$), quality of identified modular partition ($Q_{max}$), parcellation ($P$), and number of $S$-modules (identified using statistical significance based on a two-sample Wilcoxon rank-sum test) and $Q$-modules (identified based on the cutting of the dendrogram at the optimization function $Q$), (Fig. 2; Table 1; all parameters are defined in the method section). In particular, the number of $Q$-modules ranges from five to eight in non-avian archosaurs, while adult crown birds possess only two to five skull modules, showing a much higher integration of the skull (see Supplementary Data 2 file). In general, non-avian archosaurs possess a preorbital, suborbital (zygomatic arch), braincase (including the skull roof), and a left and right mandibular module (Fig. 3). In some taxa, however, the suborbital module forms a unit with the preorbital (e.g., *Archaeopteryx lithographica*, *Erlikosaurus andrewsi*) or braincase module (e.g., *Majungasaurus crenatissimus*), while in *Allosaurus fragilis* and *Gallimimus bullatus* for instance, the suborbital module is expanded, including the quadrate and squamosal. The palatal bones are either part of the preorbital or suborbital module. While the number and distribution of modules vary between species, the assignment of single bones to a certain module can vary, too, often showing a left-right asymmetry within a species. In many taxa, for instance, the premaxilla represents its own module on the one side, while it is integrated with the maxilla, nasal and lacrimal into the preorbital module on the other side (see Supplementary information; Supplementary Fig. 1).

In contrast, adult crown birds possess usually a suborbital, braincase, and a single mandibular module. The bones forming the beak are either integrated into the suborbital or braincase module. The suborbital module, however, can be independent (e.g., *Gyps fulvus*, *Milvus milvus*, *Ptychoramphus aleuticus*, *Rhea americana*) or partly integrated into the braincase module (e.g., *Bubulcus ibis*, *Phoenicopterus ruber*). In extreme cases, the skull consists of only two modules, one represented by the mandible and the other by the cranium (e.g., *Nycticorax nycticorax*, *Platalea leucorodia*). As in their ancestors, module integration of some crown birds shows a left-right asymmetry, but to a lesser degree (see Supplementary information).

All three juvenile non-avian archosaurs (*Alligator mississippiensis*, *Tarbosaurus bataar*, and *Scipionyx samniticus*) fall into the range of adult non-avian archosaurs, indicating only minor changes in bone contacts, fusion patterns, and modularity during ontogeny. Interestingly, the early juvenile theropod *S. samniticus* and the early Avialae *A. lithographica* show a very similar distribution of skull modularity (Fig. 3). Furthermore, *T. bataar* shows an ontogenetic shift in the module identity of skull roof and temporal bones, which is in concert with the fusion of the frontal bones and a reduction of the number of modules.

With the exception of connectivity ($C$), all network parameters of juvenile crown bird skulls fall between those of non-avian archosaurs and adult crown birds. Although juveniles differ significantly from the latter two groups (see Table 1), the network parameters correlate significantly with relative skull size, shifting towards the adult condition with increasing skull size (see Supplementary Fig. 2). In other words, hatchlings and early juveniles are closer to non-avian theropods/archosaurs, while subadults are closer to adult crown birds. As the same parameters (except for parameter $D$) fail to correlate with relative skull size in adult crown birds (see Supplementary Fig. 2; Table 2), these correlations represent a true ontogenetic signal. Due to their different ontogenetic growth stages, the modular organization of juvenile bird skulls is not uniform. However, some general patterns can be extracted. In many cases, premaxilla, maxilla, nasal, lacrimal, and ectethmoid/mesethmoid form a preorbital module, while the skull roof and the temporal bones form a single braincase module. The zygomatic bones (jugal and quadratojugal)

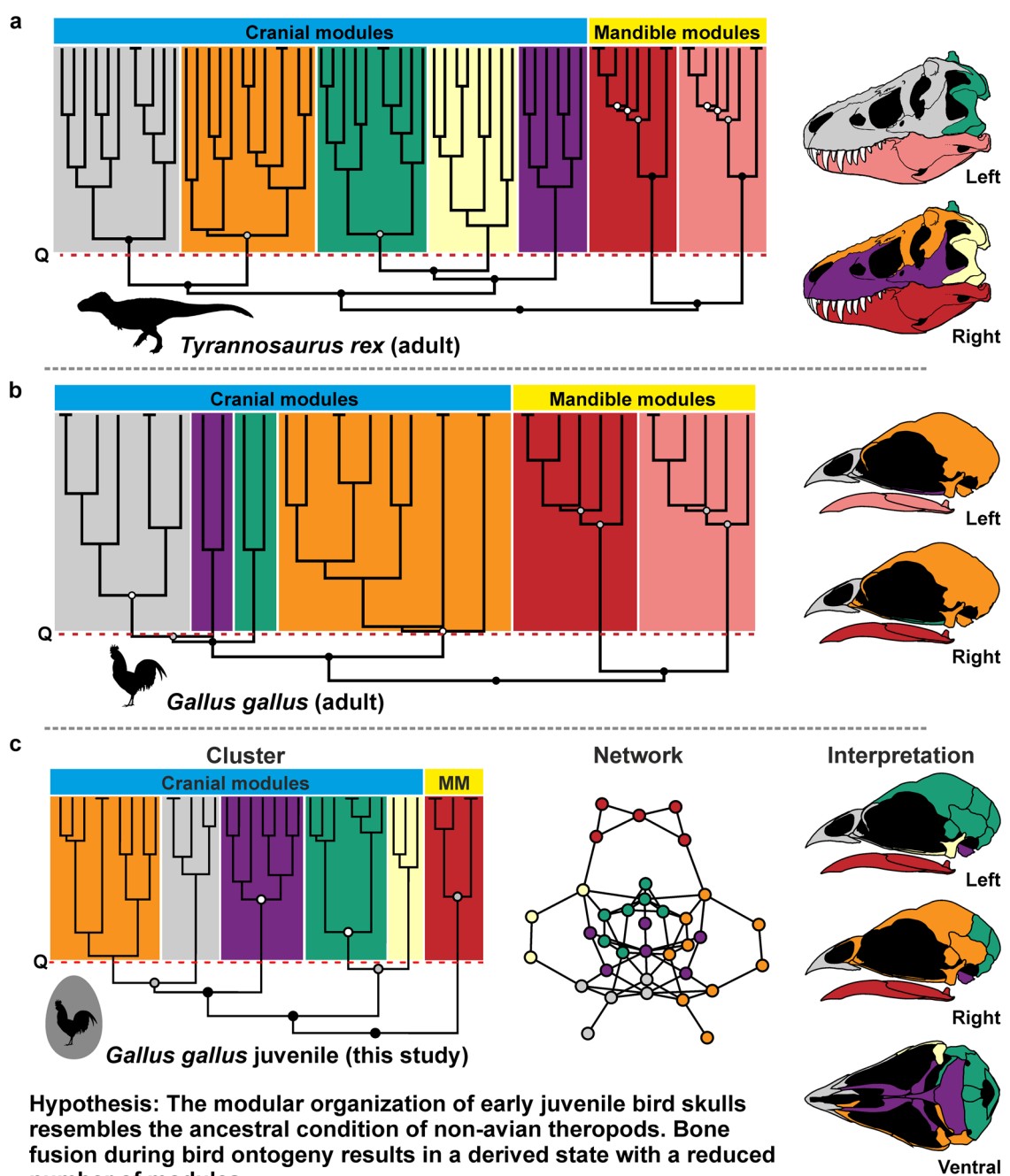

**Fig. 1 Hierarchical organization of anatomical networks in extinct and extant theropod dinosaurs. a** *UPGMA* cluster of anatomical network and skull modularity of an adult *Tyrannosaurus rex* (skull reconstruction modified from Carr[72]) modified after Werneburg et al.[11]. **b** *UPGMA* cluster of anatomical network and skull modularity of an adult *Gallus gallus* (skull reconstruction modified from Jollie[12]) modified after Werneburg et al.[11]. **c** *Ward.2D* cluster of anatomical network, anatomical network and skull modularity of a juvenile *Gallus gallus* (skull reconstruction modified from Jollie[12]) from the recent study. Horizontal dashed lines mark the partition into *Q*-modules, while circles at nodes mark the statistical significance of *S*-modules (white, *p*-value < 0.05; grey, *p*-value < 0.01; black, *p*-value < 0.001). All silhouettes are from http://www.phylopic.org/.

are often united in the suborbital module, which can also include the maxilla (*Larus ridibundus*) or the quadrate (e.g., *Pica pica*, *B. ibis*, *G. fulvus*). However, in other cases, the zygomatic bones are part of the preorbital (e.g., *Aythya ferina*, *Ciconia ciconia*) or braincase module (e.g., *Otis tarda*, *Theristicus caudatus*). In most cases, the frontal is part of the braincase module, but in some juveniles (e.g., *Gypaetus barbatus*, *Recurvirostra avosetta*, *T. caudatus*) it is part of the preorbital module or associated with the nasal and lacrimal forming a separate module (e.g., *Tyto alba*, *A. ferina*). In the latter case, the preorbital module consists of the premaxilla and is usually merged with the suborbital module. However, the frontal can also form its own module with the squamosal, laterosphenoid, quadrate, and the zygomatic bones (e.g., *Grus japonensis*, *O. tardas*). In other species, the nasal and lacrimal form with the ectethmoid/mesethmoid a second preorbital module (e.g., *Anser anser*, *Cygnus olor*). Here, the primary preorbital module is formed only by the premaxilla, maxilla, and the palatal bones. The quadrate can be linked either to the braincase (e.g., *Egretta garzetta*, *Podiceps cristatus*) or the suborbital module (e.g., *B. ibis*, *P. pica*) and the palatal bones

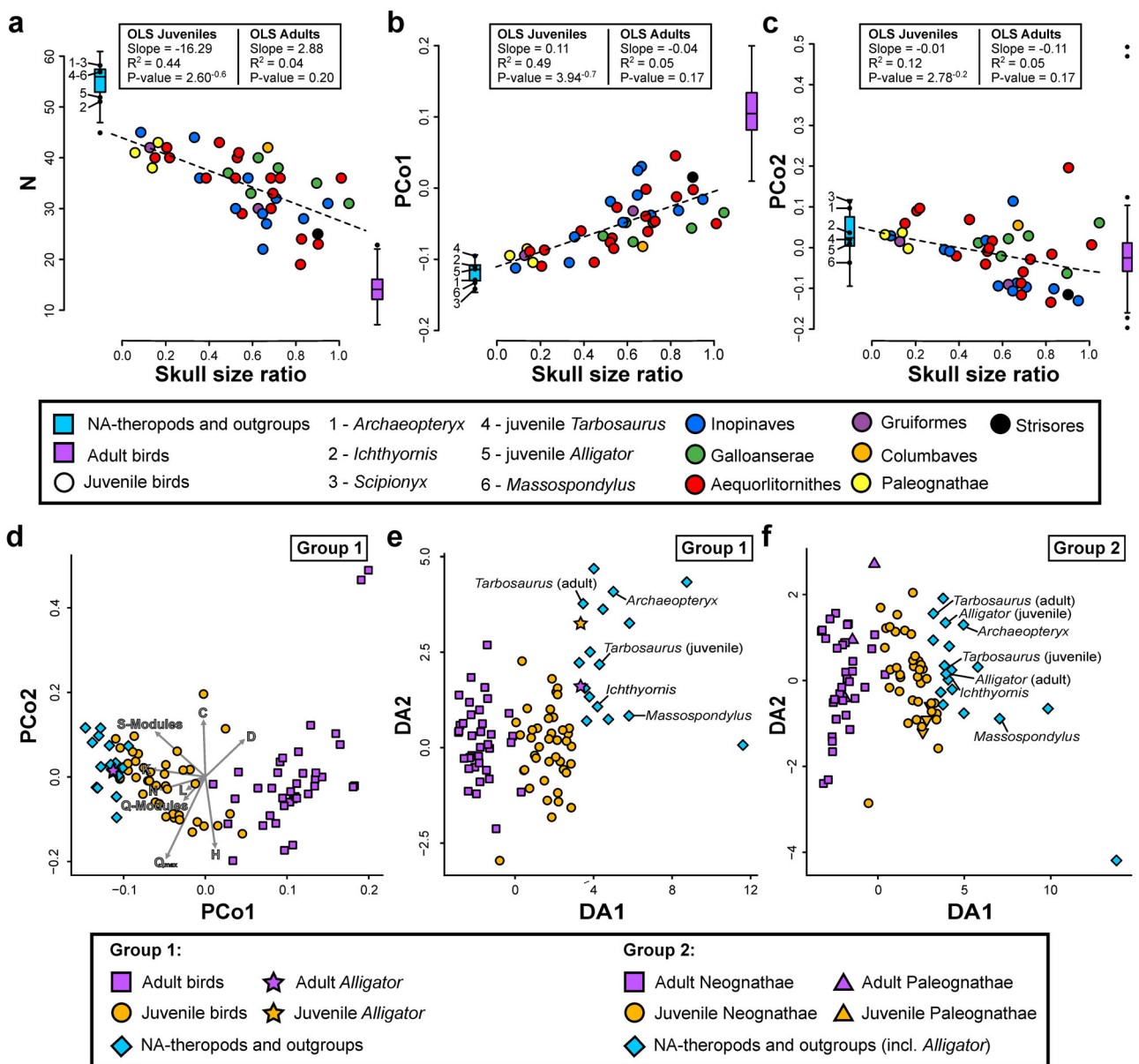

**Fig. 2 Results of the anatomical network analysis (AnNA), principal coordinate analysis (PCoA) and phylogenetic flexible discriminant analysis (pFDA). a** Range in the number of skull bones $N$ in non-avian archosaurs ($n = 19$), juvenile and adult crown birds (both $n = 41$). For juvenile crown birds, the range of $N$ is plotted against relative skull size (ratio of skull box volume). Results of ordinary least square regression analysis (OLS) describing the correlation between $N$ and relative skull size of juvenile and adult crown birds are given in the box. **b, c** Same as **a**, showing the range of PCo1 and PCo2 in non-avian archosaurs and juvenile and adult crown birds. **d** PCoA morphospace and biplot based on network parameters showing the distribution of juvenile and adult extant archosaurs and non-avian dinosaurs (Group 1). **e** pFDA plot showing the separation between juvenile and adult extant archosaurs (Group 1) and the distribution of non-avian dinosaurs. **f** pFDA plot showing the separation between juvenile and adult crown birds (Group 2) and the distribution of non-avian dinosaurs and *Alligator missippisensis*.

either form an independent module (e.g., *Ardea purpurea*) or are part of the preorbital (e.g., *Spheniscus megellanicus*), suborbital (e.g., *Upupa epops*, *P. leucorodia*), or braincase modules (e.g., *T. caudatus*). The skull networks for all specimens included in this study are illustrated in the Supplementary information (see Skull network analysis results).

**Macroevolutionary tests and ancestral state reconstruction.** The PCo1 and PCo2 of the network parameters (see principal coordinates analysis in the "Methods" section; Supplementary Data 2 file), which together account for over 75% of total variation, correlate significantly with the relative skull size of juvenile

crown birds, revealing an ontogenetic signal as described above (Fig. 2b–d). Based on the pFDA, juvenile and adult crown birds are correctly identified with an error of 2.4%, indicating a significant separation between juveniles and adults in terms of skull network structure. While most non-avian theropods resemble *Alligator* (Group 1), a comparison to juvenile and adult crown birds only (Group 2) identifies all stem-line representatives as juveniles (Fig. 2e, f; Tables 3, 4; see Supplementary Data 2 file) as indicated by PCo1 (Fig. 2d). These results are supported by the PERMANOVA: juvenile ($n = 41$) and adult birds ($n = 41$) are significantly separated from each other ($F$ value: 6.837; $p$ value: <0.001), in which the adult avian stem-line representatives ($n = 15$) are closer to *A. mississippiensis* ($n = 1$) ($F$ value: 0.284;

**Table 1 Comparison of network parameters and principal coordinates between non-avian archosaurs ($n = 19$), juvenile and adult crown birds (both $n = 41$) using the Mann–Whitney U ($z$) and the Kolmogorov–Smirnov ($D$) test.**

| | Adult birds vs. NA-archosaurs | | Juvenile birds vs. adult birds | | Juvenile birds vs. NA-archosaurs | |
|---|---|---|---|---|---|---|
| | z (p) | D (p) | z (p) | D (p) | z (p) | D (p) |
| $N$ | **6.213 (0.000)** | **1.000 (0.001)** | **7.743 (0.000)** | **0.927 (0.001)** | **6.182 (0.000)** | **0.976 (0.001)** |
| $K$ | **6.188 (0.000)** | **1.000 (0.001)** | **7.687 (0.000)** | **0.976 (0.001)** | **6.184 (0.000)** | **1.000 (0.001)** |
| $D$ | **6.185 (0.000)** | **1.000 (0.001)** | **7.579 (0.000)** | **0.902 (0.001)** | **5.651 (0.000)** | **0.853 (0.001)** |
| $C$ | 0.858 (0.395) | **0.386 (0.026)** | 0.886 (0.384) | **0.341 (0.021)** | 0.493 (0.622) | 0.225 (0.427) |
| $L$ | **6.183 (0.000)** | **1.000 (0.001)** | **7.485 (0.000)** | **0.927 (0.001)** | **5.451 (0.000)** | **0.805 (0.001)** |
| $H$ | **5.340 (0.000)** | **0.850 (0.001)** | **4.290 (0.000)** | **0.488 (0.001)** | **3.083 (0.001)** | **0.505 (0.002)** |
| $S$ | **6.859 (0.000)** | **1.000 (0.001)** | **7.476 (0.000)** | **0.805 (0.001)** | **4.513 (0.000)** | **0.537 (0.002)** |
| $Q$ | **5.948 (0.000)** | **0.902 (0.001)** | **5.750 (0.000)** | **0.659 (0.001)** | **2.501 (0.012)** | 0.266 (0.464) |
| $Q_{max}$ | **6.182 (0.000)** | **1.000 (0.001)** | **7.587 (0.000)** | **0.878 (0.001)** | **5.268 (0.000)** | **0.720 (0.001)** |
| $P$ | **5.795 (0.000)** | **0.902 (0.001)** | **6.294 (0.000)** | **0.756 (0.001)** | 1.335 (0.183) | 0.216 (0.503) |
| $PCo1$ | **6.182 (0.000)** | **1.000 (0.001)** | **7.707 (0.000)** | **0.927 (0.001)** | **5.880 (0.000)** | **0.846 (0.001)** |
| $PCo2$ | **2.972 (0.004)** | **0.517 (0.002)** | 0.751 (0.456) | 0.195 (0.310) | **2.574 (0.010)** | **0.367 (0.047)** |
| $PCo3$ | 0.779 (0.433) | 0.334 (0.085) | 1.493 (0.135) | 0.293 (0.053) | 0.509 (0.615) | 0.268 (0.262) |
| $PCo4$ | 0.334 (0.737) | 0.208 (0.511) | 0.482 (0.632) | 0.195 (0.361) | 0.111 (0.913) | 0.139 (0.931) |
| $PCo5$ | 1.701 (0.091) | 0.347 (0.077) | 0.769 (0.445) | 0.146 (0.759) | **2.559 (0.011)** | **0.420 (0.006)** |
| $PCo6$ | 0.858 (0.388) | 0.291 (0.160) | **2.105 (0.033)** | 0.268 (0.094) | **2.1463(0.015)** | **0.439 (0.008)** |
| $PCo7$ | 0.445 (0.654) | 0.209 (0.530) | 0.269 (0.787) | 0.146 (0.742) | 1.143 (0.881) | 0.212 (0.509) |
| $PCo8$ | 1.669 (0.096) | 0.343 (0.059) | 1.308 (0.0187) | 0.244 (0.182) | **2.988 (0.002)** | **0.440 (0.006)** |
| $PCo9$ | 0.222 (0.819) | 0.317 (0.108) | 0.288 (0.773) | 0.220 (0.265) | 0.175 (0.868) | 0.220 (0.454) |

P-values are given in parentheses. Tests with significant differences are shown in bold.

$p$ value: 1.00) and juvenile birds ($F$ value: 4.108; $p$ value: 0.001) than to adult birds ($F$ value: 7.935; $p$ value: <0.001).

Character evolution of $PCo1$ shows a distinct bimodal distribution for the ancestral values of stem and crown representatives with a large gap between Avialae and Aves, positioning Ornithurae (incl. *I. dispars*) closer to Aves. Such distribution indicates a severe shift in the skull network topology from Avialae towards the crown. A hypothetical substitution of adult crown birds with their juvenile counterparts (see ancestral state reconstruction in the "Methods" section) results in a more parsimonious evolution (see Supplementary Data 2 file), in which ancestral values show a unimodal distribution, reflecting a trended evolution towards the crown, but without severe shifts (Fig. 4a). Although less pronounced, the same signal is also present for $PCo2$ (Fig. 4b) and the single network parameters (see Supplementary Figs. 3, 4). Together with the previous results, this discrepancy reveals that the apomorphic skull morphology of crown birds is primarily driven by postnatal ontogeny (realized in a short period of somatic growth that is characterized by an accelerated growth rate[14,15]) and not a result of a continuous evolution along the stem lineage.

## Discussion

Besides the skull, the skeleton of adult crown birds is generally characterized by a high degree of bone fusion to the axial skeleton (e.g., cervical vertebrae with cervical ribs, neural spines of the dorsal vertebrae forming a notarium, the synsacrum, and the pygostyle) and to the appendicular skeleton (e.g., the scapulo-coracoid, carpometacarpus, pelvic bones, and the tarsometatarsus)[16,17]. Tracing the evolutionary history of bone fusion in the manus and pelvis along maniraptoran theropods indicates that these features evolved many times independently across non-avialan theropods, Enantiornithes, and Ornithuromorpha, while bone fusion events within different body regions are uncoupled from each other and furthermore do not correlated with the growth pattern, i.e. neither duration nor speed of somatic growth impact bone fusion patterns[18].

While the skull network of early juvenile crown bird skulls recapitulates the condition of adult non-avian theropods, the intensive bone fusion during bird ontogeny exaggerates the ancestral adult traits of non-avian theropods, leading to a stronger modular integration in terms of connectivity. Interestingly, this ontogenetic pattern is paralleled to a lesser degree in *Tarbosaurus*, where the fusion of the frontal bones causes a reduction in the number of modules. Because bone fusion generally requires a previous contact of bones in form of a suture, it represents the more derived state with respect to the suture, both at the ontogenetic[13,19] and evolutionary level[18,20]. Therefore, we hypothesize that the intense fusion in the skull ontogeny of crown birds represents a peramorphic heterochrony (i.e., the developmental exaggeration of the ancestral adult trait) with respect to their non-avian theropod ancestors that evolved very late during avian evolution, at the origin of crown-birds.

Heterochrony was previously proposed to play a key role in theropod skull evolution leading to the avian skull[7,8,21,22]. A sequence of paedomorphic events in coelurosaurs, for instance, manifested juvenile traits of more basal theropods in the shape of the maxilla, nasal, orbit and brain of adult birds[7,8,21]. However, the relative increase of the premaxilla in Pygostylia, for instance, was already identified as localized peramorphosis[7]. In this regard, our finding highlights that the origin of bird skulls was much more complex, including opposite heterochronies acting at different levels (i.e., overall shape, local shape and bone network), but also at different phylogenetic entities.

While the initial phase of paedomorphosis in avian skull evolution is probably related to a trend of body size reduction in coelurosaurs[7,21,23] caused by the truncation of the growth period[14,15], we speculate that the peramorphic bone fusion in crown birds is related to the origin of cranial kinesis. Finite-element-analyses on extant bird skulls indicate that the fused areas of the premaxillae and the braincases are regions of low strain, while most stress acts on the mobile nasal-frontal hinge and the zygomatic arch[24,25], which are the key components in avian skull kinesis[2]. Thus, we speculate that the fusion-induced immobility of the braincase and the premaxillary tip constrains a

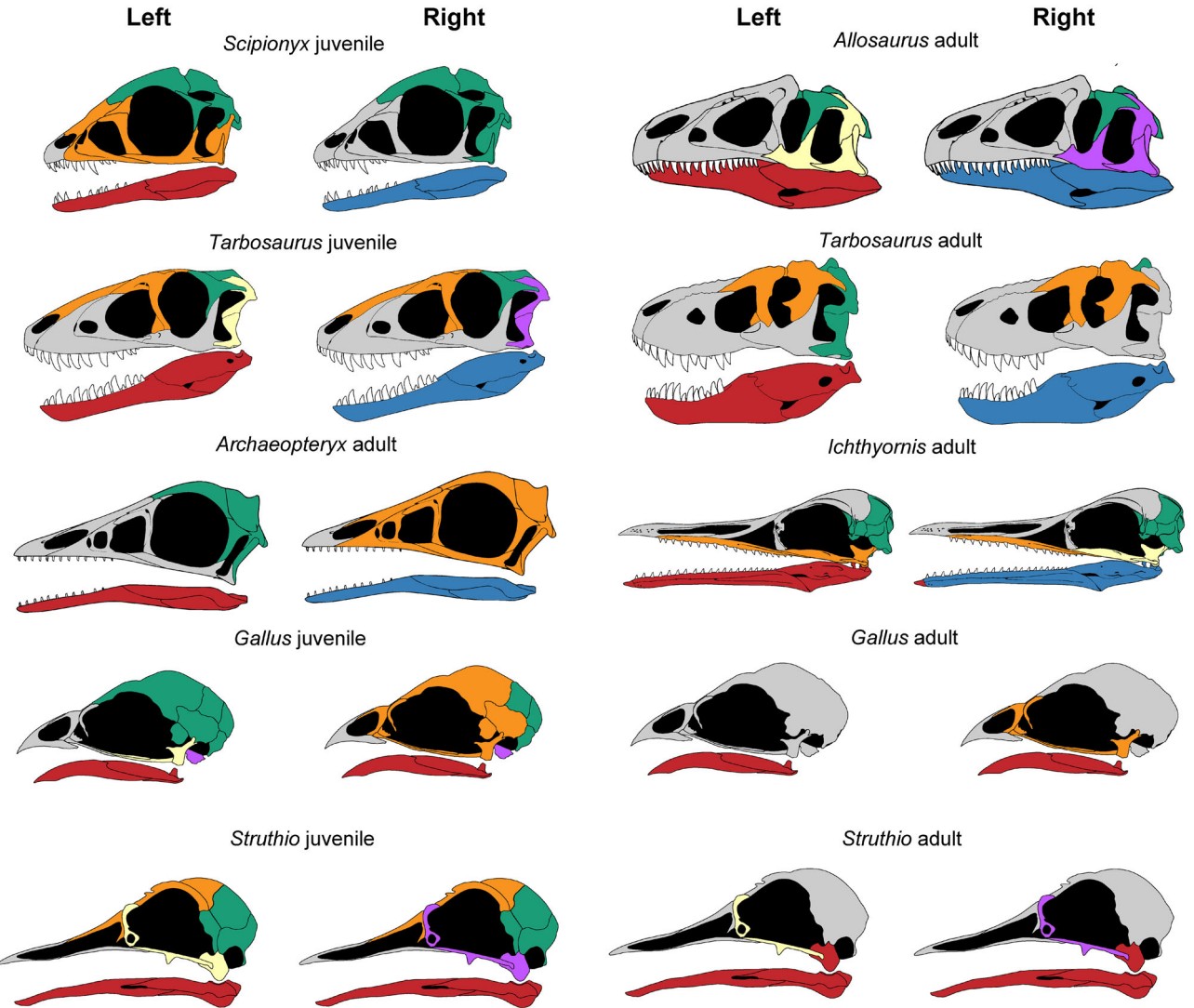

**Fig. 3 Distribution of skull modules in some non-avian theropods, and ontogenetic pairs of *Gallus gallus* and *Struthio camelus* from the left and right (mirrored) side.** The modularity of non-avian theropods is very similar between different species (including non-avian Avialae), but also throughout ontogeny (see ontogenetic pair of *Tarbosaurus bataar*). Juvenile crown birds resemble the ancestral condition in number and distribution of modules, while adult crown birds show a severe reduction of modules due to ontogenetic bone fusion. Colours highlight different skull modules within species, but do not necessarily imply homology between species. Original sources of modified skull reconstruction: *Scipionyx samniticus*[73], *Allosaurus fragilis*[1], juvenile *Tarbosaurus bataar*[74], adult *Tarbosaurus bataar*[75], *Archaeopteryx lithographica*[5], *Ichthyornis dispar*[6], *Gallus gallus*[12], *Struthio camelus*[76] (see Supplementary Data 2 file).

controlled kinetic dorsoventral movement of the avian beak during biting/picking.

Although resembling juvenile theropods in shape[7] (Fig. 3), our study confirms that *Archaeopteryx lithographica* possesses a rather theropod-like skull, lacking cranial kinesis[5,26]. This akinetic condition is also present in *Confuciusornis sanctus*, *Sapeornis chaoyangensis* and basal Enantiornithes[4,27,28], although the reduction of the jugal-postorbital bar in some Enantiornithes may indicate primitive kinesis in some species[4]. As indicated by the palatal morphology and the lack of a jugal-postorbital bar, the first definite evidence for simple cranial kinesis is present in Late Cretaceous Ornithurae like *Ichthyornis dispars* and *Hesperornis regalis*[6,29]. Although the frontal-parietal suture is still open, *I. dispars* shows a partial fusion between the premaxillae, frontals and possibly the parietals[30]. Thus, the fossil record supports the hypothesis that the late origin of cranial kinesis and bone fusion in the history of birds may be linked. However, the skull network of *I. dispars* is still different from adult crown birds and resembles

the condition of juveniles instead. In contrast, phylogenetic reconstructions of the palatal morphology[21] and cranial network (this study) indicate that the last common ancestor of grown birds had a strongly fused skull with a kinetic motion. While palaeognaths still have an intermediate stage, the most advanced skull kinesis is realized in neognath birds[28]. If skull bone fusion is further related with the short, accelerated growth mode of crown birds still need to be tested, but examples from other body regions indicate that bone fusion and growth strategy are not necessarily linked with each other[18].

Beside the ontogenetic signal in network parameters, many taxa in our sample, including non-avian theropods, show a left-right asymmetry in modularity, which is caused when paired bones are assigned to different modules or unpaired bones do not form their own module, but are assigned either to a left or right module. Similar asymmetries were previously found in skull network analyses of amniote and synapsid skulls[11,31] and the skull muscularity of primates and deformed humans[32,33]. On the

**Table 2 Comparison of correlations of network parameters and principal coordinates against relative skull size (ratio of skull box volume) in juvenile and adult crown birds (both $n = 41$).**

| | Juvenile birds | | Adult birds | | ANCOVA |
|---|---|---|---|---|---|
| | **Slope** | **$R^2$ ($p$)** | **slope** | **$R^2$ ($p$)** | **$F$ ($p$)** |
| $N$ | **−16.290** | **0.436 (0.000)** | 2.881 | 0.041 (0.204) | **25.930 (0.000)** |
| $K$ | **−37.930** | **0.454 (0.000)** | 4.985 | 0.030 (0.280) | **27.380 (0.000)** |
| $D$ | **0.054** | **0.258 (0.001)** | −0.092 | **0.058 (0.120)** | **6.123 (0.002)** |
| $C$ | 0.018 | 0.005 (0.6720) | −0.052 | 0.009 (0.570) | 0.538 (0.466) |
| $L$ | **−0.690** | **0.347 (0.000)** | 0.187 | 0.026 (0.317) | **13.830 (0.000)** |
| $H$ | **0.165** | **0.297 (0.000)** | 0.055 | 0.021 (0.363) | 2.363 (0.128) |
| $S$ | **−2.179** | **0.348 (0.000)** | −0.149 | 0.006 (0.620) | **12.44 (0.000)** |
| $Q$ | **−2.068** | **0.286 (0.000)** | 1.086 | 0.076 (0.084) | **15.550 (0.000)** |
| $Q_{max}$ | **−0.130** | **0.325 (0.000)** | 0.060 | 0.033 (0.253) | **10.300 (0.002)** |
| $P$ | **−0.092** | **0.221 (0.002)** | 0.115 | 0.082 (0.073) | **9.892 (0.002)** |
| PCo1 | **0.106** | **0.487 (0.000)** | −0.039 | 0.047 (0.174) | **19.570 (0.000)** |
| PCo2 | **−0.095** | **0.118 (0.028)** | −0.112 | 0.045 (0.172) | 0.035 (0.852) |
| PCo3 | −0.010 | 0.091 (0.056) | 0.037 | 0.006 (0.642) | 0.243 (0.138) |
| PCo4 | 0.038 | 0.013 (0.475) | −0.097 | 0.040 (0.206) | 2.125 (0.149) |
| PCo5 | −0.035 | 0.010 (0.516) | 0.051 | 0.011 (0.499) | 0.894 (0.347) |
| PCo6 | −0.029 | 0.006 (0.617) | −0.081 | 0.043 (0.183) | 0.369 (0.545) |
| PCo7 | −0.082 | 0.042 (0.200) | −0.006 | 0.000 (0.933) | 0.715 (0.400) |
| PCo8 | −0.036 | 0.011 (0.511) | −0.099 | 0.052 (0.151) | 0.535 (0.467) |
| PCo9 | −0.020 | 0.003 (0.748) | −0.018 | 0.002 (0.813) | 0.000 (0.983) |

Using ordinary least square regression analysis (OLS), the correlation between trait and size is shown by the coefficient of determination $R^2$ and a $p$-value for the slope. The equality of slopes for the juvenile and adult regression was tested with ANCOCA. Tests with significant differences are shown in bold.

**Table 3 Results of the phylogenetic flexible discriminant analysis (pFDA) based on the PCoA data, comparing the separation between juvenile and adult crown birds and *Alligator mississippiensis* (Group 1).**

| Training data (extant archosaurs) | **AA** | **AJ** | **CA** | **CJ** | |
|---|---|---|---|---|---|
| Aves-Adult ($n = 41$; AA) | 40.000 | 1.000 | 0.000 | 0.000 | |
| Aves-Juvenile ($n = 41$; AJ) | 1.000 | 40.000 | 0.000 | 0.000 | |
| Crocodylia-Adult ($n = 1$; CA) | 0.000 | 0.000 | 1.000 | 0.000 | |
| Crocodylia-Juvenile ($n = 1$; CJ) | 0.000 | 0.000 | 0.000 | 1.000 | |
| *Error of identification* | *0.024* | *0.024* | *0.000* | *0.000* | |

| Test data (extinct archosaurs) | **P(AA)** | **P(AJ)** | **P(CA)** | **P(CJ)** | **Prediction** |
|---|---|---|---|---|---|
| *Massospondylus* | 0.000 | 0.040 | **0.950** | 0.010 | Crocodylia-Adult |
| *Herrerasaurus* | 0.000 | 0.000 | 0.010 | **0.990** | Crocodylia-Juvenile |
| *Coelophysis* | 0.000 | 0.000 | **0.965** | 0.035 | Crocodylia-Adult |
| *Majungasaurus* | 0.000 | 0.195 | **0.796** | 0.009 | Crocodylia-Adult |
| *Sinraptor* | 0.000 | 0.000 | 0.000 | **1.000** | Crocodylia-Juvenile |
| *Allosaurus* | 0.000 | 0.000 | **1.000** | 0.000 | Crocodylia-Adult |
| *Acrocanthosaurus* | 0.000 | 0.000 | 0.016 | **0.984** | Crocodylia-Juvenile |
| *Tarbosaurus* (adult) | 0.000 | 0.009 | 0.123 | **0.868** | Crocodylia-Juvenile |
| *Tarbosaurus* (juvenile) | 0.000 | 0.002 | **0.998** | 0.000 | Crocodylia-Adult |
| *Gallimimus* | 0.000 | **0.549** | 0.406 | 0.045 | **Aves-Juvenile** |
| *Erlikosaurus* | 0.000 | 0.218 | **0.774** | 0.008 | Crocodylia-Adult |
| *Citipati* | 0.000 | 0.385 | 0.071 | **0.544** | Crocodylia-Juvenile |
| *Velociraptor* | 0.000 | 0.007 | 0.000 | **0.993** | Crocodylia-Juvenile |
| *Gobivenator* | 0.000 | **0.793** | 0.200 | 0.007 | Aves-Juvenile |
| *Archaeopteryx* | 0.000 | 0.000 | 0.000 | **1.000** | Crocodylia-Juvenile |
| *Ichthyornis* | 0.000 | 0.133 | **0.863** | 0.004 | Crocodylia-Adult |

The phylogenetic strength $\lambda$ is 0.01 and the error for the correct identification of extant taxa to their original group is 0.024. Non-avian dinosaurs were mostly identified as juvenile or adult *Alligator mississippiensis*. Only the ornithomimosaur *Gallimimus* and the troodontid *Gobivenator* were identified as close to juvenile crown birds. Values in italics highlight the group specific error of identification. Values in bold highlight the predicted group based on the highest probability.

one hand, this could be a methodological artefact caused by the dichotomous nature of the cluster analysis, which is not able to produce fully symmetric topologies when paired bones possess an equal probability of being part of one module or another[34]. This effect is amplified by the presence of non-paired elements (e.g., basisphenoid, basioccipital, supraoccipital), which impose an artificial asymmetry onto the network, as the cluster algorithm cannot decide whether these elements are linked to the left or the right. To test this, we deleted all unpaired elements from the dataset of *A. lithographica* and repeated the anatomical network analysis. This modification results in a more or less symmetrical cluster, where the distribution of modules for *A. lithographica* is

**Table 4 Results of the phylogenetic flexible discriminant analysis (pFDA) based on the PCoA data, comparing the separation between juvenile and adult crown birds (Group 2).**

| Training data (extant birds) | NeoA | NeoJ | PA | PJ | |
|---|---|---|---|---|---|
| Neognathae-Adult (n = 38; NeoA) | 37.000 | 1.000 | 2.000 | 0.000 | |
| Neognathae-Juvenile (n = 38; NeoJ) | 1.000 | 36.000 | 0.000 | 3.000 | |
| Palaeognathae-Adult (n = 3; PA) | 0.000 | 0.000 | 1.000 | 0.000 | |
| Palaeognathae-Juvenile (n = 3; PJ) | 0.000 | 1.000 | 0.000 | 0.000 | |
| *Error of identification* | *0.026* | *0.053* | *0.667* | *1.000* | |

| Test data (extinct archosaurs + *Alligator*) | P(NeoA) | P(NeoJ) | P(PA) | P(PJ) | Prediction |
|---|---|---|---|---|---|
| *Alligator* (adult) | 0.000 | **0.787** | 0.000 | 0.213 | Aves-Juvenile (NeoJ) |
| *Alligator* (juvenile) | 0.000 | **0.967** | 0.000 | 0.033 | Aves-Juvenile (NeoJ) |
| *Massospondylus* | 0.000 | 0.009 | 0.000 | **0.991** | Aves-Juvenile (PJ) |
| *Herrerasaurus* | 0.000 | 0.000 | 0.000 | **1.000** | Aves-Juvenile (PJ) |
| *Coelophysis* | 0.000 | 0.003 | 0.000 | **0.997** | Aves-Juvenile (PJ) |
| *Majungasaurus* | 0.000 | **0.858** | 0.000 | 0.142 | Aves-Juvenile (NeoJ) |
| *Sinraptor* | 0.000 | **0.994** | 0.000 | 0.006 | Aves-Juvenile (NeoJ) |
| *Allosaurus* | 0.000 | **0.687** | 0.000 | 0.313 | Aves-Juvenile (NeoJ) |
| *Acrocanthosaurus* | 0.000 | **0.990** | 0.000 | 0.010 | Aves-Juvenile (NeoJ) |
| *Tarbosaurus* (adult) | 0.000 | **0.991** | 0.000 | 0.009 | Aves-Juvenile (NeoJ) |
| *Tarbosaurus* (juvenile) | 0.000 | **0.825** | 0.000 | 0.175 | Aves-Juvenile (NeoJ) |
| *Gallimimus* | 0.000 | 0.369 | 0.000 | **0.631** | Aves-Juvenile (PJ) |
| *Erlikosaurus* | 0.000 | **0.660** | 0.000 | 0.340 | Aves-Juvenile (NeoJ) |
| *Citipati* | 0.000 | **0.973** | 0.000 | 0.027 | Aves-Juvenile (NeoJ) |
| *Velociraptor* | 0.000 | **0.905** | 0.000 | 0.095 | Aves-Juvenile (NeoJ) |
| *Gobivenator* | 0.000 | **0.590** | 0.000 | 0.410 | Aves-Juvenile (NeoJ) |
| *Archaeopteryx* | 0.000 | **0.941** | 0.000 | 0.059 | Aves-Juvenile (NeoJ) |
| *Ichthyornis* | 0.000 | **0.764** | 0.000 | 0.236 | Aves-Juvenile (NeoJ) |

The phylogenetic strength $\lambda$ is 0.11 and the error for the correct identification of extant taxa to their original group is 0.098. Non-avian dinosaurs were always identified as juvenile birds. Values in italics highlight the group specific error of identification. Values in bold highlight the predicted group based on the highest probability.

identical on the left and right side (see Supplementary information). However, Powell et al.[32] described an increase of module asymmetry within the head and neck muscles of simiiform primates, which they interpreted to be related to more complex, asymmetrical facial expressions. Diogo et al.[33] further found that developmental deformation during embryogenesis increase module asymmetry. These two examples indicate that left-right asymmetry may not entirely be a methodological artefact, but could have a true biological meaning. Thus, it could be possible that the left-right asymmetry results from developmental (only for juveniles) and biomechanical (for juveniles and adults) constraints, indicating a more complex modular hierarchy that cannot be resolved with the current methodology. This has to be tested in the future studies in more detail using different methodologies for calculating modularity, like *OSLOM* (Order Statistics Local Optimization Method)[34,35]. Nevertheless, as all network parameters apart from the connectivity (C) show the same ontogenetic and evolutionary trend (see Supplementary Figs. 2, 3), modular asymmetry apparently has no impact on our interpretation of the results (see Supplementary Fig. 5, see Supplementary Data 2 file).

In summary, our study demonstrates that the final step towards highly modular integrated skulls evolved in the last common ancestor of the bird crown, caused by rather abrupt peramorphic bone fusion at the origin of crown-birds that adds to the underlying skull shape paedomorphosis in the evolution of coelurosaurs, highlighting the mosaic evolution of the bird skull. Further events of opposite heterochronies along the stem-line of birds may have occurred in dental evolution. The tooth morphology of small-bodied adult coelurosaurs, like dromaeosaurids and compsognathids, is probably paedomorphic with respect to their ancestors, as it resembles that of juvenile basal tetanurans[36]. In contrast, the ontogenetic increase in the number of teeth in troodontids probably represents a peramorphic heterochrony[37].

Oppositional heterochronies can be further found in the growth evolution of birds, in which the somatic growth of crown birds is characterized by a short duration, but high speed[14,15]. When compared with the mode of their ancestors[38,39], the former growth pattern represents a progenetic paedomorphosis, while the latter is an accelerated peramorphosis[40,41]. Finally, oppositional heterochronies have been further suggested for arthropod heads[42] and human skull evolution[43]. Together with recent embryological studies on the artificial induction of ancestral snout and tooth morphologies[44,45] and the verification of temporary ossification centres of bones that are absent in hatched birds[46,47], our study shows that the genotype and the phenotypic development of crown birds still contains relics from their theropod ancestors, while modifications in the timing, location and intensity of developmental processes result in the evolutionary novelty that is the avian skull.

## Methods

**Sampling.** The sampling includes 41 extant birds for which both juvenile and adult specimens were available for study (see Supplementary Data 2 file). Juvenile specimens were identified as such based on the presence of skull sutures on the skull roof, which is a reliable proxy for determining maturity in birds[13]. Although skeletal material of early juvenile birds is generally limited in osteological collections, species representing all major bird lineages could be sampled, but due to fast skeletal growth, the exact age of juvenile birds cannot usually be determined. As a consequence, the juveniles sampled herein do not necessarily represent the same ontogenetic stage (i.e., hatchlings or subadults). Skulls of adult birds are much more frequent in osteological collections and can be identified easily as such by means of their high degrees of bone fusion in the skull. Thus, each bird species is represented by an ontogenetic pair, including a juvenile and an adult.

In addition, we sampled the skulls of twelve non-avialan theropod dinosaur species, two avialan stem birds (*Archaeopteryx lithographica* and *Ichthyornis dispars*) and two outgroup taxa (*Massospondylus carinatus* and *Alligator mississipiensis*). Finally, to explore ontogenetic modularity in non-avian archosaurs, we added three juvenile specimens: *Scipionyx samniticus*, *Tarborausurus bataar* and *Alligator mississipiensis* (see Supplementary Data 2 file). Because AnNA, which sources the contact/non-contact of biological structures, requires complete

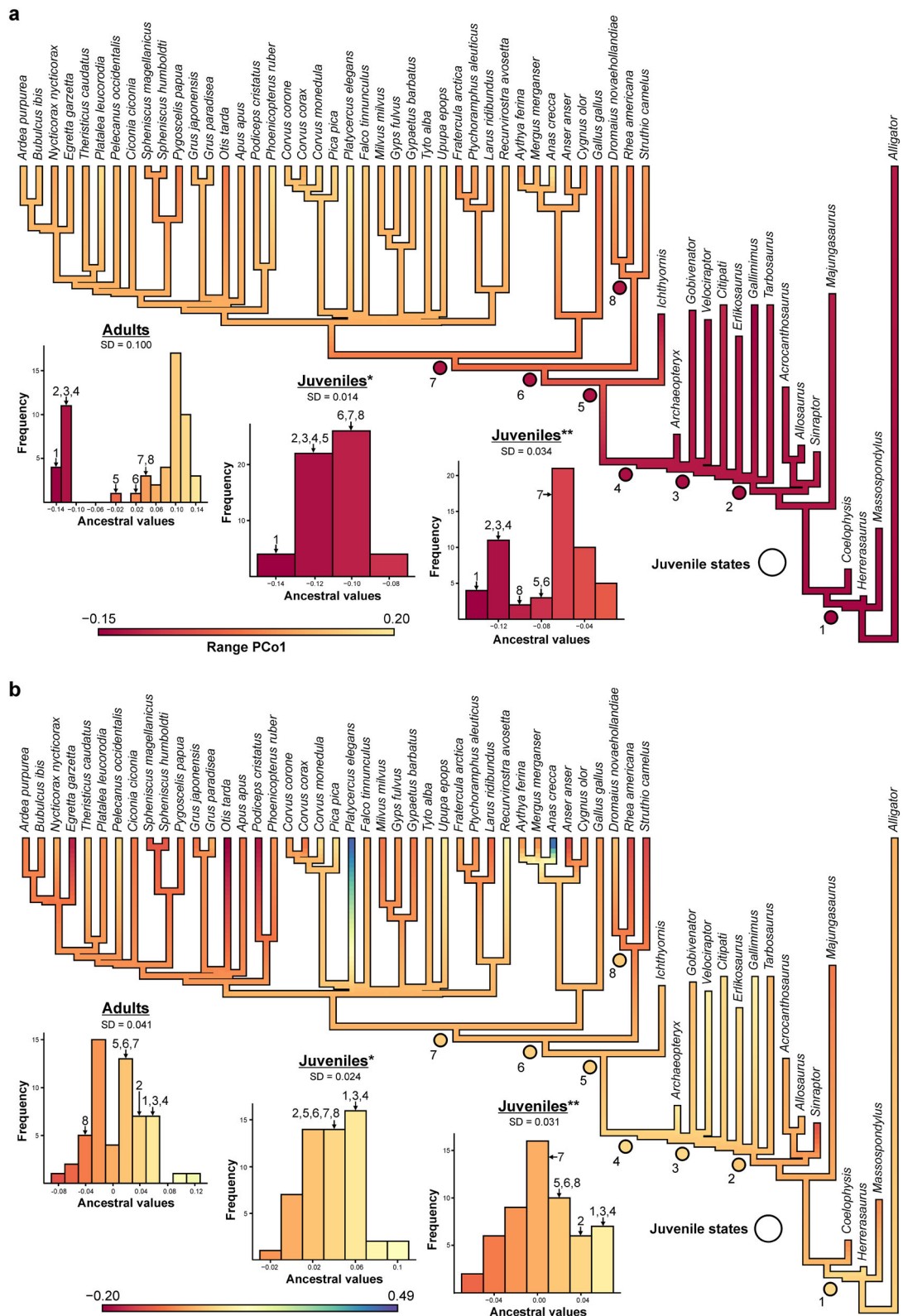

**Fig. 4 Ancestral state reconstruction of PCoA data for adult and juvenile crown birds. a** *PCo1.* **b** *PCo2.* Based on ancestral values, skull network evolution shows a severe shift from Avialae (4) to Ornithurae (5) and Aves (6) as indicated in the histogram. Substituting adult crown birds with juveniles (see circles; histogram: *residual values; **original values) results in a more parsimonious evolution (see standard deviation SD), where the change from Avialae (4) to Aves (6) is rather continuous (see Supplementary Data 2 file). (1) Theropoda; (2) Coelurosauria; (3) Eumaniraptora; (4) Avialae; (5) Ornithurae; (6) Aves; (7) Neognathae; (8) Palaeognathae.

skulls, where each bone contact can be reconstructed, only a small fraction of fossil stem-line representatives could be sampled. This is especially true for juvenile specimens, which have a smaller likelihood of preservation and discovery[48].

All crown-group birds were scored based on first-hand observations, while the scoring of the stem-line representatives and outgroup taxa sources from a combination of first-hand observations and the scientific literature (see Supplementary Data 2 file).

**Anatomical network analysis.** Based on the recent publication of Werneburg et al.[11], we used AnNA to analyse the skulls in our sample in terms of modularity. The anatomical network matrix is based on a symmetric unweighted and undirected adjacency matrix[34], where the nodes ($N$) represent bones and connections ($K$) represent the presence of a link between two bones. Bones that are clearly separated from each other by a suture (e.g., the suture between the frontal and parietal) or represent an articulation (e.g., the articulation between the quadrate and articular) were treated as independent units. In contrast, fused bones were treated as one unit (see Esteve-Altava et al.[20]). All bones or units that are not linked with each other were scored as "0" whereas sutures or articulations between bones or units were scored as "1" (see Supplementary Data 1 file).

Following the script of Werneburg et al.[11], the data matrix of each specimen was analysed with the help of the software R-3.5.2[49] and the package igraph[50]. These analyses determined the following network parameter for each specimen: number of nodes ($N$), of connections ($K$), density of connections ($D$), mean clustering coefficient ($C$), mean shorted path length ($L$) and variance of connectivity ($H$)[51]. Here, parameters $N$ and $K$ represent the number of bones and their physical contacts with each other, respectively. $D$ measures the existing number of connections ($K$) divided by the maximum number of connections possible. $C$ is the arithmetic mean of the ratio of a node's neighbours that connect among them in a triangular manner. $L$ measures the average of the shortest path length between all pairs of bones. $H$ is the standard deviation and the mean of the number of connections of all nodes in the network[51,52]. Modules were identified by the hierarchical clustering of the generalized topological overlap similarity matrix among nodes ($GTOM$), generated from the primary data matrices of each species, which assumes that nodes connecting to the same other nodes (i.e., shared neighbours) more likely belong to the same module. However, in contrast to Werneburg et al.[11] we did not use $UPGMA$, but $Ward.D2$ for cluster analysis as this method minimizes variance[53,54]. The number of modules was determined from the clusters using the optimization function modularity $Q$[55], which evaluates if the modules identified are better supported than what is expected at random. Accordingly, these modules are called $Q$-modules, in which the quality of the identified modular partition is expressed by the parameter $Q_{max}$, which quantifies the number of connections between nodes within the same module minus the expected number of connections if they were distributed at random among the same modules for the best of the possible partitions. Furthermore, we estimated $S$-modules, which are based on a two-sample Wilcoxon rank-sum test on the internal vs. external connections of each module. In addition, we also calculated the parcellation ($P$), which measures the extent to which the nodes of a network are divided into balanced modules. The theoretical background of these module analyses are described, among others, in Werneburg et al.[11], Rasskin-Gutman and Esteve-Altava[51], Esteve-Altava et al.[52,56] and Clauset et al.[55].

**Statistics and reproducibility.** For comparing the range of each network parameter between juvenile and adult crown birds (both $n = 41$) and the outgroup taxa (i.e., non-avian theropods, *Massospondylus carinatus* and *Alligator mississippiensis*; $n = 19$) (see Supplementary Fig. 2), the nonparametric Mann–Whitney $U$ and Kolmogorov–Smirnov tests were applied, which are more robust against the non-normal distribution of data[57]. The Mann–Whitney $U$ test calculates whether two univariate samples are taken from populations with equal medians, while the Kolmogorov–Smirnov test compares whether the shape of two univariate distributions is similar or not[57].

As the juveniles do not have the same ontogenetic stage (see above), the network parameters were plotted against relative skull size (not total size), which is expressed as the percentage of the adult bird skull box volume (length × width × height) for each species, respectively. By standardizing size this way, it is possible to document how the network parameters change during growth. Correlations between network parameters and relative skull size were tested with ordinary least square (OLS) regression analysis. To assure that these correlations represent a true ontogenetic, and not just an allometric signal, the same correlations were tested for adult birds. Here, relative size was expressed as the percentage of the skull box volume of the largest bird (*Pelecanus occidentalis*) sampled. The equality of the regression slopes of both analyses were compared to each other using a one-way ANCOVA based on an F test in *PAST v. 3.05*[58].

Next, we log-transformed the resulting network parameters ($N$, $K$, $D$, $C$, $L$, $H$, $S$-Modules, $Q$-Modules, $Q_{max}$) and applied PCoA using the Gower index[59,60], which is the default measure for using mixed data types (see Supplementary Data 2 file). Similar to principal component analysis (PCA), PCoA reduces a multivariate dataset down to a small set of dimensions (principal coordinates, PCo's) associated with a measure of the variance (eigenvalue) for each *PCo*, and allows comparing the distribution of juvenile and adult birds with each other and with respect to their non-avian ancestors and *Alligator mississippiensis* in a multivariate space. In order

to test for an ontogenetic/allometric signal, we performed an OLS and one-way ANCOVA between the single *PCo*'s and relative skull size for both juvenile and adult birds as described above.

Two time-calibrated supertrees were created, which served as the phylogenetic framework for various statistical analyses and character evolution analyses. The supertrees differ from each other in terms of the crown-group topology, as one is based on that of Hackett et al.[61], the other on that of Ericson et al.[62]. To assess temporal uncertainty, we downloaded a set of 1000 relaxed-clock trees for each topology from the webpage birdtree.org[63,64], which summarize the range of uncertainties in terms of time calibration of ancestral nodes from molecular clock estimations. From those trees, we computed a temporal consensus for each topology, using the function *consensus.edges* in the *phytools* package[65] of R. The final topologies were completed by the addition of the sampled stem line representatives and *Alligator mississippiensis* following the general consensus on non-avian theropod phylogeny[1,66] (see Supplementary Data 3 file). To explore how differences in the crown bird topology can affect the outcome, all phylogenetic-based methods (see below) were run with both supertrees and compared with each other, while the presentation of the results is based on the topology of Hackett et al.[61].

To test for the statistical overlap between the different ontogenetic groups in the crown (i.e., juvenile vs. adult) and their relation to stem line representatives in the multivariate morphospace (see PCoA), we applied a phylogenetic flexible discriminant analyses (pFDA)[67,68] in R. This version of classical discriminant analyses first estimates Pagel's lambda testing how the grouping correlates with phylogeny, and then applies this assessment for controlling for phylogenetic non-independence during the actual discriminant analyses. To apply this method for ontogenetic series, all crown-group birds in the supertree were split into two OTUs (one representing the juvenile and the other the adult specimen), each having a branch length of one year. Although the ontogenetic growth of birds is not equal, this value had to be standardized, as pFDA requires an isometric tree. The same was done for the ontogenetic pairs of *Alligator mississippiensis* and *Tarbosaurus bataar*, each with a branch length of ten years, taking the longer ontogenetic growth of both species into account when compared to birds. Afterwards, we divided the extant OTUs into four groups: (1) juvenile and (2) adult birds (both $n = 41$), and (3) juvenile and (4) adult *Alligator mississippiensis* (both $n = 1$). pFDA tests if the extant groups can be separated from each other or not, and assigns all fossil taxa to the groups based on their original position in the morphospace. In a second run, we divided only birds into four groups: (1) juvenile and (2) adult Palaeognathae (both $n = 3$), and (3) juvenile and (4) adult Neognathae (both $n = 38$). The degree of overlap between the different groups was additionally tested using a permutational multivariate analysis of variance (PERMANOVA)[69] in *PAST*. The PERMANOVA was run with 10,000 replications, Euclidean distance as the distance measure and Bonferroni correction, in which the $p$ values were multiplied with the number of comparisons to decreases the impact of multiplicity and the probability of rejecting incorrectly the null hypothesis[70].

Finally, we explored how skull modularity changed through bird evolution. Using a maximum likelihood-based ancestral state reconstruction with a Brownian motion model having a constant rate of diffusion, $PCo1$ and $PCo2$ of adult birds, non-avian dinosaurs and *Alligator mississippiensis* were mapped onto the two supertrees. This was done using *Ace* function in the R package *Ape*[71]. As this ancestral state reconstruction reflects the true evolution for adult semaphoronts, we wanted to further know, how much of this evolution is actually affected by bird ontogeny. Therefore, we repeated the analysis by substituting the adult birds with their juvenile counterparts. However, as the juvenile birds in our sample do not represent the same ontogenetic stages, trait changes along the branches are potentially affected by ontogenetic signals included in the data. To reduce this effect, we estimated the residuals from the OLS regression of $PCo1$ and $PCo2$ against relative skull size and standardized these values with the youngest individual in our sample (*Struthio camelus*), transforming all juveniles to hypothetical hatchlings (see Supplementary Data 2 file). For comparing the trait evolution of both analyses with each other, we estimated the standard deviation from all ancestral values, which quantifies the amount of variation and can be used as a proxy for parsimony. Likewise, parsimony was determined by the log-transformed product of all ancestral values. Finally, we compared the ancestral values of selected clades with each other, including Theropoda, Coelurosauria, Eumaniraptora, Avialae, Ornithurae, Aves, Neognathae and Palaeognathae.

**Reporting summary.** Further information on research design is available in the Nature Research Reporting Summary linked to this article.

## Data availability
All data for the anatomical network analyses is part of the Supplementary information (Supplementary Data 1 file).

## Code availability
The R code for the anatomical network analyses is part of the Supplementary information (Supplementary Data 4 file). The R code for the phylogenetic flexible discriminant analysis are provided by Schmitz and Motani[67].

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

## Acknowledgements

We thank Michel Beaud (Musée d'Histoire Naturelle Fribourg), Christine Lefèvre (Museum National d'Histoire Naturelle Paris), Hans-Martin Berg (Naturhistorisches Museum Wien), Manuel Schweizer and Reto Hagmann (Naturhistorisches Museum Bern) for access to their ornithological collections. Walter Joyce, Serjoscha Evers, Felix Quade and Kévin Le Verger are thanked for discussions and comments on the manuscript. This study was funded by the Swiss National Science Foundation (PZ00P2_174040) and the German Science Foundation (FO 1005/2-1) (to C.F.).

## Author contributions

C.F. designed the research project; O.P. and C.F. collected and analysed the data; and O.P. and C.F. wrote the paper and prepared all figures.

## Competing interests

The authors declare no competing interests.
