## [Peer Review File · Communications Biology]

Reviewers' comments:

Reviewer #1 (Remarks to the Author):

The authors present a broad comparative study of the avian anatomy. They used a novel method, anatomical network analysis, to address how avian skull development evolved different forms. The study reports the presence of developmental trends in the morphological complexity, integration, and modularity of the skulls that match the evolutionary transformations seen from non-avian theropods to modern birds. The authors propose the fusion of bones as the leading mechanisms driving this evolution-development parallelism of heterochrony.

The study has a broad appeal to general audiences because of its broad focus on avian skull evo-devo, and presents interesting new results in the context of recent advances in morphological sciences. Overall, the study is well designed and methodologically sound. I have only a few methodological comments and minor suggestions that can be easily addressed by the authors at this point. These are necessary in my opinion, but they do not invalidate the work already done, which is impressive, especially for the amount and diversity of data gathered.

Major comments:

(1) I do not agree with the use of Q_{\max} as a measure of strength of modularity. I acknowledge that this type of interpretation may have been used before, but it is wrong. The reason is that Q_{\max} is calculated for a particular network (and its randomization) and a community detection algorithm, therefore it should only be used to compare the output of different algorithms on a same network. Not to compare different networks. A higher or lower Q_{\max} between two different networks is not informative of which is more or less modular. To put it in another way (and differences aside), it is like comparing two p-values of a same statistical test for two independent samples and groupings. Because it is not a comparable measure among different networks, Q_{\max} should neither be included in following analyses nor as one of the variables for PCA (or similar methods).

As a way to measure the strength of modularity, and compare it among skulls, I propose to use instead the parameter parcellation (as described in doi 10.1126/sciadv.aau7459 or doi 10.1111/evo.13430).

(2) The authors report the presence of modularity asymmetries in some skulls. In many instances this is a consequence of the method used to identify connectivity modules. The reason is that many methods (like the one used but also many others) do not take into account the presence of singletons (nodes not assigned to any module) or the overlap of modules (nodes belonging to two or more modules at the same time), which are very reasonable biological expectations. To check if this is the case of the skulls having asymmetry, I suggest to use a complementary method to identify modules: OSLOM (Order Statistics Local Optimization Method), as described in doi 10.1371/journal.pone.0018961. It has been shown that combining the method used by the authors together with OSLOM allows to better interpret the modular organization of anatomical networks and yields more robust results (see doi 10.1002/jmor.20690).

(3) Authors use a phylogenetic flexible discriminant analysis to test statically the overlap or separation among groups from PCA scores (lines 88-89 and 308-324). I am not sure if this is the best approach. Because FDA is not an actual hypothesis tests (I may be wrong), to assess whether PC scores discriminate between groups I would suggest, for example, using a PERMANOVA (as in doi 10.1126/sciadv.aau7459).

(4) Regression between network parameters and skull size ratio are made using OLS. I wonder if it would be more informative to perform pGLS in juvenile and adults separately. Does the OLSs performed take into account phylogenetic relations?

Minor comments:

I think a figure showing the output of the modularity analysis (they are as supplementary figures) can benefit the impact of the article, as it visually shows the results. At least a couple of key examples. This will also help readers new to network analysis to see how a skull network looks like.

The formatting of the manuscript is a bit confusing and does not help the authors to make their point (e.g. the references are splitted between sections, lack of headings and sub-headings). I would suggest a more classical format.

The comments (1 to 6) on results and methods that appear in the supplementary would improve a revised discussion section. I would suggest integrate them in the discussion.

Line 51. "To test this hypothesis". It would help the reader to state the hypothesis explicitly. It is not clear enough from the previous lines what is the specific hypothesis.

Line 57 [paragraph starting at]. The authors start presenting the results on network parameters, but it comes a bit hard for readers not familiar with these parameters. I would suggest adding an introductory paragraph linking the above hypothesis testing with the network parameters (and their morphological interpretation) to help readers transition to the bulk of the results. Authors can find a review of these parameters in the already cited articles.

This review is not anonymous.

Signed, Borja Esteve-Altava

Reviewer #2 (Remarks to the Author):

This work presents a new take on the evolutionary trends of the avian skull analyzing the network structure of the skull in different points of the ontogeny, something previously not done.

The number of species and phylogenetic range of the species used seems adequate and the results are interesting in light of the previous results regarding the subject

I have one major consideration and a point to question, I will start with that one first.

In line 37 you claim the jaws of birds are capable of independent simultaneous movement, please clarify this. The upper jaw can or not move with respect to the braincase depending on the morphology and articulation of the quadrate and palate elements, but the jaw elements are fused and the independent movement of both sides is restricted.

My major concern about the article is the asymmetric networks obtained and their significance. When establishing a network, medial, non-paired or fused elements can impose an artificial asymmetry on the network as the algorithms working can not decide whether this node is more connected to the left side or the right, "choosing" one or the other. I imagine this might extend the asymmetry to elements that are strongly connected to these nodes and expand the asymmetry. I recommend paying attention to this and either revise the effect of the programs used for establishing the networks in generating these asymmetries or discuss in depth if the possible biological meaning they could have if you believe they do represent a true anatomical property of the network.

A recent paper published last year (Dinosaur ossification centres in embryonic birds uncover developmental evolution of the skull) discusses the embryonic presence of "lost" elements in the avian skull. Although your samples are post-embryonic, the results of this paper seem in line with your results and the idea that either embryonic or juvenile skulls seem to be more complex than adult ones, and more similar to ancestral forms.

Also, it highlights that some in some theropod dinosaurs adult independent bones had a tendency to fuse, late in ontogeny. This fusion tendency might be exaggerated in birds leading to their skull being almost completely fused.

Also, a couple of papers by Shuo Wang show that in the some dinosaurs teeth were lost during ontogeny, leading to an edentulous beak. Heterochrony and acceleration of these pattern could be linked and relate to your results, and could also be discussed

Reviewer #3 (Remarks to the Author):

The authors present a novel and concise report that contributes to our knowledge of avian evolution. It is among the few articles that studied network modularity in birds, prominently including ontogenetic data. It is also mostly congruent with what we know about avian skull evolution, including the more integrated anatomical networks of modern birds compared with other archosaurs (Werneburg et al., 2019; Esteve-Altava, doctoral dissertation (available online: <http://roderic.uv.es/handle/10550/31487>); or Esteve-Altava et al., 2013; surprisingly uncited in the current work). However, there are some major issues with the current version of the paper. Amending these problems that I suggest below involves reframing the study. A revised re-framed version of this manuscript should be suitable for publication in *Communications Biology*.

One of the main important issues I see is that the authors use indistinctly two concepts that are very different: variational and connectivity modularity/integration. Variational and connectivity modularity are two different things (the first, describes the variational hierarchy of traits, often shape in the context of GM, the other describes the hierarchy of connections) and the interplay between both is not understood at all in anatomical systems (there is not any published article to my knowledge on this issue). For instance, in lines 42-47 the authors contrast results from variational modularity in modern birds with results from connectivity modularity in a theropod dinosaur. These comparisons are meaningless as different sets of bones could be varying jointly in shape during ontogeny but not be connected among them until much later in development. This is the case of the brain and the facial bones, which co-vary in morphology as a result of the effect of several morphogenetic factors but are not connected through bony structures until much later in development (e.g., Hu et al., 2015).

The second important issue directly stems from the first, and it is related with the heterochronic claims the authors made. Heterochrony (including its outcomes, like peramorphosis) involves evolutionary (or intraspecific for some authors) changes in the timing of events during ontogeny. In the quantitative formalization that most of the community uses and accepts currently (i.e., Alberch et al., 1979) the concept alludes to the variations in shape relative to variation in size (often as a proxy of ontogenetic time). The authors claim that ontogenetic variation in several anatomical network variables over ontogeny suggest that evolutionary variation in the lineage leading to the crown could have been acquired via an heterochronic process. But this loose definition is tricky and it is very dependent on the variable selected, for instance, if you select Mean clustering coefficient (C) or PCo2 scores it does not change in a very directional way over ontogeny or evolution. Also, it is expected that paedomorphosis will result in earlier fusion of bones, loses (or apparent loses, see Smith-Paredes et al., 2018) and these effects would result in a more integrated (less modules) and more dense

networks (higher density of connections) (Esteve-Altava et al., 2013). In this case, the main ontogenetic event that varies between avians and non-avians is the fusion of bones, which happens earlier and more profusely. This together with shape changes in the overall morphology has the side effect of getting groups of bones together (i.e., reduction of the antorbital fenestra) and therefore the physical connections between bones are easier. Therefore, on the light of this evidence, the results the authors present are what one would expect from a structure undergoing an heterochronic process resulting in a paedomorphic shape, which is very interesting in its own right. As a result, I would not recommend using the term peramorphosis to describe the network changes over ontogeny and evolution as it implies that the evolutionary changes in skull network configuration results from an extension (or acceleration) of ontogenetic change which is likely not the case (and probably on the light of direct developmental evidence, it is likely the opposite).

Also, on a more technical side: number of connections is expected to vary together with number of bones but a measure of that is not a measure of complexity (e.g., Esteve-Altava et al., 2014). Furthermore, density of connections could be interpreted as a measure of skull complexity and it increases significantly in adult birds as compared with juveniles and non-avians, therefore all references to a decrease in complexity in modern birds should be changed. This pattern coincides with previous work on the effect of Willinston's Law, in which previous authors identified that bone fusions and losses are coupled with an increase in density of connections, and they interpreted as a way of development to cope with the harmful effects of these re-organizations (Esteve-Altava et al., 2013).

In fact, in my opinion, this paper should be restructured to stress that: 1) it advances our knowledge on the evolution of anatomical networks in birds over ontogeny and evolution, this is interesting and useful in its own right; also, 2) because it describes the same system that has been tied by previous convincing work to particular developmental process producing paedomorphosis, it represents a rare opportunity to test how those heterochronic changes affect connectivity patterns of involved structures; and finally, 3) the restructuring role of development in coping with bone losses and fusions (Esteve-Altava et al., 2013) and its connection with cranial kinesis and kinesis complexity in birds should be discussed more in depth. In fact, I think that the expected role of paedomorphic process in anatomical networks (bone fusions and losses), the balancing effect of development (augmenting connectivity between the remaining bones) and the role of kinesis (selecting those complex associations between groups of bones for functional roles) all point the same direction and could explain your results much better than the current unfounded claims of peramorphism.

Other minor issues involve the discussion of results from variational modularity and integration. Although Felice & Goswami, 2017 reported a modular organization of variation in shape in the skull of birds in a follow up paper Felice et al., 2019 found that the skull of modern birds is more integrated than non-avian dinosaurs and crocodylians. There are also a number of studies in selected groups of birds that point towards an integrated skull (Bright et al., 2016, 2019; in diurnal raptors and parrots respectively). However, there are still important issues with the quantification of modularity and integration in geometric morphometrics (see e.g., Cardini, 2019) which are likely to be affecting variation in shape in the skull of birds and because there are not ideal solutions yet, these results should be taken cautiously.

Minor comments:

Lines 7-11: this phrase should be deleted or simplified 'Triggered by numerous fossil discoveries from the Late Jurassic and Early Cretaceous of China, the evolutionary transition from terrestrial non-avian theropods to volant birds has become a vital, interdisciplinary research field, often focusing on feather evolution and the origin of flight.' I think it detracts from the main message of the paper.

Line 13: strong modular integration is ambiguous. You should state if you mean to variational or connectivity modularity/integration.

Lines 32, 33: change 'aberrant' for 'modified'

Line 57: 'including the stem-birds, Archaeopterys and Ichthyornis '

Line 59: The authors should explain the difference between Q-modules and S-modules, one statement should be fine in here as the rest is explained in the Methods section.

Lines 218-219: This phrase should be deleted as it is not its place in the Methods section:

'Ontogenetic changes in skull modularity of birds were investigated in a macroevolutionary context.'

Bibliography: Revise the spelling of 'Borja Esteve-Altava' and various other authors.

Figures:

GENERAL: Although I do not consider this critical for publication of the paper I wanted to suggest a series of new visualizations that might improve the paper:

1) Biplots in the PCoA plots so we can see how particular variables account for how variation is distributed in the multidimensional space. Also, it will be great to have a PCo1 vs PCo3 plot and a PCo2 vs PCo3 plot, with overlain biplots.

2) A composite figure showing the relationship of, at least, the most important pairs of variables derived from the anatomical (for instance number of bones against density of connections). If the authors are concerned about not showing the role of size at the same time, they can plot it as symbol size very easily using functions within ggplot2, for instance. Plotting a phylogeny over these plots will be desirable, particularly as the authors have composite phylogenies with all the specimens including juveniles.

Figure 1: To improve clarity I suggest adding silhouettes to the plots d, e, f coloured as the three groups they represent. Also, because DFA PCoA vectors' direction is arbitrary I suggest changing one or the others so the distribution of the groups in the plot is the same in the three.

Figure 2: I suggest drawing the palatine bones and portraying the ventral/dorsal aspects of a more reduced set of taxa. I understand that this might be difficult, an alternative solution to this might be placing a the composite figure of pairs of network variables I suggested above instead.

Bibliography:

Esteve-Altava, B., Marugán-Lobón, J., Botella, H. and Rasskin-Gutman, D., 2013. Structural constraints in the evolution of the tetrapod skull complexity: Williston's law revisited using network models. *Evolutionary Biology*, 40(2), pp.209-219.

Esteve-Altava, B., Marugán-Lobón, J., Botella, H. and Rasskin-Gutman, D., 2014. Random loss and selective fusion of bones originate morphological complexity trends in tetrapod skull networks. *Evolutionary Biology*, 41(1), pp.52-61.

Felice, R.N., Watanabe, A., Cuff, A.R., Noirault, E., Pol, D., Witmer, L.M., Norell, M.A., O'Connor, P.M. and Goswami, A., 2019. Evolutionary integration and modularity in the archosaur cranium. *Integrative and comparative biology*.

Hu, D., Young, N.M., Xu, Q., Jamniczky, H., Green, R.M., Mio, W., Marcucio, R.S. and Hallgrímsson, B., 2015. Signals from the brain induce variation in avian facial shape. *Developmental Dynamics*, 244(9), pp.1133-1143.

Smith-Paredes, D., Núñez-León, D., Soto-Acuña, S., O'Connor, J., Botelho, J.F. and Vargas, A.O., 2018. Dinosaur ossification centres in embryonic birds uncover developmental evolution of the skull. *Nature ecology & evolution*, 2(12), p.1966.

Reviewer #1 (Remarks to the Author):

The authors present a broad comparative study of the avian anatomy. They used a novel method, anatomical network analysis, to address how avian skull development evolved different forms. The study reports the presence of developmental trends in the morphological complexity, integration, and modularity of the skulls that match the evolutionary transformations seen from non-avian theropods to modern birds. The authors propose the fusion of bones as the leading mechanisms driving this evolution-development parallelism of heterochrony.

The study has a broad appeal to general audiences because of its broad focus on avian skull evo-devo, and presents interesting new results in the context of recent advances in morphological sciences. Overall, the study is well designed and methodologically sound. I have only a few methodological comments and minor suggestions that can be easily addressed by the authors at this point. These are necessary in my opinion, but they do not invalidate the work already done, which is impressive, especially for the amount and diversity of data gathered.

- **We are thankful for the general assessment of our study.**

Major comments:

(1) I do not agree with the use of Q_{max} as a measure of strength of modularity. I acknowledge that this type of interpretation may have been used before, but it is wrong. The reason is that Q_{max} is calculated for a particular network (and its randomization) and a community detection algorithm, therefore it should only be used to compare the output of different algorithms on a same network. Not to compare different networks. A higher or lower Q_{max} between two different networks is not informative of which is more or less modular. To put it in another way (and differences aside), it is like comparing two p-values of a same statistical test for two independent samples and groupings. Because it is not a comparable measure among different networks, Q_{max} should neither be included in following analyses nor as one of the variables for PCA (or similar methods).

- **In the previous version of the manuscript, we followed the definition of Q_{max} provided by the reviewer in 2014. Based on his criticisms, we redefined it as a measure for the quality of identified modular partition (see Werneburg et al. 2019).**
- **Q_{max} was originally included into the PCoA, because it showed an ontogenetic signal, in which the quality of modular partition is higher in those specimens that have a higher number of modules.**

As a way to measure the strength of modularity, and compare it among skulls, I propose to use instead the parameter parcellation (as described in doi 10.1126/sciadv.aau7459 or doi 10.1111/evo.13430).

- **We are thankful for this suggestion. Unfortunately, this parameter was not introduced in Werneburg et al. 2019 (which was also co-authored by the reviewer). Thus, we are a little surprised that only Q_{max} was included in the script provided by Werneburg et al. 2019, but not the parcellation (P). Nevertheless, we integrated the parameter P into our data set. As for the other parameters, we tested the correlation with relative skull size and character evolution.**
- **In order to test if the inclusion/exclusion of Q_{max}/P has any impact on the results of the PCoA (as pointed out by the reviewer), we repeated the analysis three-times (with Q_{max} and P excluded; with only P included and with both parameters included) and compared it with our original PCoA. The inclusion/exclusion of both parameters has no impact on the**

general outcome of the PCoA, as indicated by the figure and regression scores in the table below. Based on the biplots, *P* and *Q*_{max} point more or less in the same direction (towards negative values of PCo2), in which the vector of *Q*_{max} is approximately twice as long as *P*. Because of the general similarity of the results and the fact the heterochronic signal is primarily included in PCo1, we did not update the PCoA.

PCo1	Slope	R	P-Value
PCo1(+Qmax;-P) vs PCo1(-Qmax;-P)	0.97	0.998	2.16E-16
PCo1(+Qmax;-P) vs PCo1(-Qmax;+P)	0.99	0.992	2.16E-16
PCo1(+Qmax;-P) vs PCo1(+Qmax;+P)	1.01	0.994	2.16E-16
PCo2	Slope	R	P-Value
PCo2(+Qmax;-P) vs PCo2(-Qmax;-P)	0.84	0.746	2.16E-16
PCo2(+Qmax;-P) vs PCo2(-Qmax;+P)	0.94	0.866	2.16E-16
PCo2(+Qmax;-P) vs PCo2(+Qmax;+P)	0.93	0.997	2.16E-16

(2) The authors report the presence of modularity asymmetries in some skulls. In many instances this is a consequence of the method used to identify connectivity modules. The reason is that many methods (like the one used but also many others) do not take into account the presence of singletons (nodes not assigned to any module) or the overlap of modules (nodes belonging to two or more modules at the same time), which are very reasonable biological expectations. To check if this is the case of the skulls having asymmetry, I suggest to use a complementary method to identify

modules: OSLOM (Order Statistics Local Optimization Method), as described in doi 10.1371/journal.pone.0018961. It has been shown that combining the method used by the authors together with OSLOM allows to better interpret the modular organization of anatomical networks and yields more robust results (see doi 10.1002/jmor.20690).

- **The protocol of Werneburg et al. does not include OSLM, although the problem of asymmetric modules was also present (see occipital region of *Tyrannosaurus*). Nevertheless, we downloaded the program and tried to run some of our datasets. Unfortunately, the program is not very intuitive and without a proper introduction, we feel that the interpretation of the results risk to be insubstantial. Instead, we already discussed the problem of asymmetry in the supplementary information and shifted this section into the main text.**

(3) Authors use a phylogenetic flexible discriminant analysis to test statically the overlap or separation among groups from PCA scores (lines 88-89 and 308-324). I am not sure if this is the best approach. Because FDA is not an actual hypothesis tests (I may be wrong), to assess whether PC scores discriminate between groups I would suggest, for example, using a PERMANOVA (as in doi 10.1126/sciadv.aau7459).

- **pFDA is a test of an actual hypothesis as one defines groups a priori and tests if single data points can be assigned correctly to this grouping. The difference between MANOVAs and DAs is that MANOVAS test if the medians/means of different groups overlap or not, while DAs try to enforce a separation of the groups. Therefore, we do not agree that pFDA represents a wrong test for our question. However, as PERMANOVA produces very similar results, we are happy to include the results into our study.**

(4) Regression between network parameters and skull size ratio are made using OLS. I wonder if it would be more informative to perform pGLS in juvenile and adults separately. Does the OLSs performed take into account phylogenetic relations?

- **To avoid a misunderstanding, the network parameters of juvenile and adult birds were separately tested for correlation with skull size ratio. However, the OLS is not phylogenetically corrected. According to Hennig, phylogenetic comparisons have to be done based on semaphoronts, i.e. individuals of different species with the same ontogenetic stage. As the juvenile individuals in our sample do not fulfil this criterion (some of them are early juveniles/hatchlings; others are subadults), the comparison between network parameters and relative skull size would create artificial variation. This is especially enforced between sister species with highly different ontogenetic stages. Using semaphoronts for phylogenetic comparison, for instance, is also the reason, why we estimated size corrected residuals for PCo1 and PCo2, transforming them to hypothetical hatchlings.**

Minor comments:

I think a figure showing the output of the modularity analysis (they are as supplementary figures) can benefit the impact of the article, as it visually shows the results. At least a couple of key examples. This will also help readers new to network analysis to see how a skull network looks like.

- **We added a figure (now Figure 1) that shows how networks, clusters and modularity are related to each other using juvenile *Gallus* as example.**

The formatting of the manuscript is a bit confusing and does not help the authors to make their point (e.g. the references are splitted between sections, lack of headings and sub-headings). I would suggest a more classical format.

- **The manuscript was originally formatted and submitted as a Letter to Nature. For the revision, we adapted the text to the format of Communications Biology.**

The comments (1 to 6) on results and methods that appear in the supplementary would improve a revised discussion section. I would suggest integrate them in the discussion.

- **Major parts of the supplementary information were included into the main text, but to preserve the flow of the discussion some points on the robustness and different modularity analyses remain in the supplementary information.**

Line 51. "To test this hypothesis". It would help the reader to state the hypothesis explicitly. It is not clear enough from the previous lines what is the specific hypothesis.

- **We specified the hypothesis in the text and stated it also in Fig. 1.**

Line 57 [paragraph starting at]. The authors start presenting the results on network parameters, but it comes a bit hard for readers not familiar with these parameters. I would suggest adding an introductory paragraph linking the above hypothesis testing with the network parameters (and their morphological interpretation) to help readers transition to the bulk of the results. Authors can find a review of these parameters in the already cited articles.

- **We are thankful for this suggestion and extended the method section explaining the meaning of the parameters in more detail. Due to the given manuscript structure of the journal, the method is unfortunately located at the end of the text. This makes a review of the parameter at the beginning of the result section problematic in terms of repetition. Therefore, we added a short comment where to find the details of the parameters.**

This review is not anonymous.

Signed, Borja Esteve-Altava

Reviewer #2 (Remarks to the Author):

This work presents a new take on the evolutionary trends of the avian skull analyzing the network structure of the skull in different points of the ontogeny, something previously not done. The number of species and phylogenetic range of the species used seems adequate and the results are interesting in light of the previous results regarding the subject.

- **As for reviewer1, we are thankful for the general assessment of our study.**

I have one major consideration and a point to question, I will start with that one first.

In line 37 you claim the jaws of birds are capable of independent simultaneous movement, please clarify this. The upper jaw can or not move with respect to the braincase depending on the morphology and articulation of the quadrate and palate elements, but the jaw elements are fused and the independent movement of both sides is restricted.

- **Due word limitation for our initial submission, we have kept this section brief. We extended/clarified this statement for the revised version.**

My major concern about the article is the asymmetric networks obtained and their significance. When establishing a network, medial, non-paired or fused elements can impose an artificial asymmetry on the network as the algorithms working can not decide whether this node is more connected to the left side or the right, "choosing" one or the other. I imagine this might extend the asymmetry to elements that are strongly connected to these nodes and expand the asymmetry. I recommend paying attention to this and either revise the effect of the programs used for establishing the networks in generating these asymmetries or discuss in depth if the possible biological meaning they could have if you believe they do represent a true anatomical property of the network.

- **We added and extended a short discussion on the modular asymmetry in the discussion of the main text, which was previously in the supplementary information.**

A recent paper published last year (Dinosaur ossification centres in embryonic birds uncover developmental evolution of the skull) discusses the embryonic presence of "lost" elements in the avian skull. Although your samples are post-embryonic, the results of this paper seem in line with your results and the idea that either embryonic or juvenile skulls seem to be more complex than adult ones, and more similar to ancestral forms. Also, it highlights that some in some theropod dinosaurs adult independent bones had a tendency to fuse, late in ontogeny. This fusion tendency might be exaggerated in birds leading to their skull being almost completely fused.

- **We are thankful for this suggestion. Actually, we had considered this study for our discussion, but due to reference limitations for our initial submission, it was not included in the final version. However, as Communications Biology allows for more citations, we are happy to include this reference again.**

Also, a couple of papers by Shuo Wang show that in the some dinosaurs teeth were lost during ontogeny, leading to an edentulous beak. Heterochrony and acceleration of these pattern could be linked and relate to your results, and could also be discussed

- **We are happy to add this study to our discussion.**

Reviewer #3 (Remarks to the Author):

The authors present a novel and concise report that contributes to our knowledge of avian evolution. It is among the few articles that studied network modularity in birds, prominently including ontogenetic data. It is also mostly congruent with what we know about avian skull evolution, including the more integrated anatomical networks of modern birds compared with other archosaurs (Werneburg et al., 2019; Esteve-Altava, doctoral dissertation (available online: <http://roderic.uv.es/handle/10550/31487>); or Esteve-Altava et al., 2013; surprisingly uncited in the current work). However, there are some major issues with the current version of the paper. Amending these problems that I suggest below involves reframing the study. A revised re-framed version of this manuscript should be suitable for publication in Communications Biology.

- **The reason why some references are not cited has been given above. We are happy to include the relevant references suggested by the reviewer.**
- **We do not agree with the interpretation of the results by the reviewer and thus do not consider reframing our study. The specific reasons are given below.**

One of the main important issues I see is that the authors use indistinctly two concepts that are very different: variational and connectivity modularity/integration. Variational and connectivity modularity are two different things (the first, describes the variational hierarchy of traits, often shape in the context of GM, the other describes the hierarchy of connections) and the interplay between both is not understood at all in anatomical systems (there is not any published article to my knowledge on this issue). For instance, in lines 42-47 the authors contrast results from variational modularity in modern birds with results from connectivity modularity in a theropod dinosaur. These comparisons are meaningless as different sets of bones could be varying jointly in shape during ontogeny but not be connected among them until much later in development. This is the case of the brain and the facial bones, which co-vary in morphology as a result of the effect of several morphogenetic factors but are not connected through bony structures until much later in development (e.g., Hu et al., 2015).

- **We do not agree that we mixed these concepts. We just briefly summarized results of studies that deals with modularity in birds and theropods, pointing out that they based on different approaches. We even included a short section in the supplementary information, discussing the different outcomes of both approaches. The analytical part of our study, of course, relies only on connectivity modularity. Nevertheless, we rephrased the section to avoid confusion.**

The second important issue directly stems from the first, and it is related with the heterochronic claims the authors made. Heterochrony (including its outcomes, like peramorphosis) involves evolutionary (or intraspecific for some authors) changes in the timing of events during ontogeny. In the quantitative formalization that most of the community uses and accepts currently (i.e., Alberch et al., 1979) the concept alludes to the variations in shape relative to variation in size (often as a proxy of ontogenetic time). The authors claim that ontogenetic variation in several anatomical network variables over ontogeny suggest that evolutionary variation in the lineage leading to the crown could have been acquired via an heterochronic process. But this loose definition is tricky and it is very dependent on the variable selected, for instance, if you select Mean clustering coefficient (C) or PCo2 scores it does not change in a very directional way over ontogeny or evolution.

- **This statement somehow implies that we were cherry picking to enforce a heterochronic signal. We do not agree with this allegation. As indicated by our regression analyses PCo2 has a significant ontogenetic signal and the absence of such signal is only true for the network parameter C.**
- **The reviewer should note that this ontogenetic signal is present in 9 out of 10 (as P is now included, see reviewer1) parameters. In order to avoid analysing 9 times the same effect, we performed a PCoA that summarized the ontogenetic effects as two variables, PCo1 (69.4%) and PCo2 (6.2%), which both summarize about 75% of total variation. In contrast, the remaining 25% that have no ontogenetic signal is spread over 7 PCo's, of which PCo3 summarize less than 5%.**

Also, it is expected that paedomorphosis will result in earlier fusion of bones, loses (or apparent loses, see Smith-Paredes et al., 2018) and these effects would result in a more integrated (less modules) and more dense networks (higher density of connections) (Esteve-Altava et al., 2013). In this case, the main ontogenetic event that varies between avians and non-avians is the fusion of bones, which happens earlier and more profusely. This together with shape changes in the overall morphology has the side effect of getting groups of bones together (i.e., reduction of the antorbital fenestra) and therefore the physical connections between bones are easier. Therefore, on the light of this evidence, the results the authors present are what one would expect from a structure

undergoing an heterochronic process resulting in a paedomorphic shape, which is very interesting in its own right. As a result, I would not recommend using the term peramorphosis to describe the network changes over ontogeny and evolution as it implies that the evolutionary changes in skull network configuration results from an extension (or acceleration) of ontogenetic change which is likely not the case (and probably on the light of direct developmental evidence, it is likely the opposite).

- **We do not agree with this statement at all. First of all, we never question the paedomorphic evolution of the bird skulls in terms of shape evolution. In fact, we fundamentally agree with this, and one of the authors (CF) has even published an article (Foth et al. 2016) demonstrating that the shape paedomorphosis in the stem-line of birds goes even deeper than suggested by Bhullar et al. (2012) (modern birds still have an antorbital fenestra, it is just not fully enclosed, but connected with the orbit. We guess the reviewer meant size reduction).**
- **Second, peramorphosis (see Alberch, Gould, Klingenberg, etc.) is present, if the adult trait of a descendent is more derived than in the ancestor, while the juvenile condition of the descendent still resembles the ancestral condition. This is exactly what we see in our data. The juvenile birds resemble the ancestral patterns of non-avian theropods, while the fused skulls of adult birds represent the derived state (not present in the ancestors), in which the derived state is defined by the ontogenetic trajectory. The reviewer, actually states this it herself: "In this case, the main ontogenetic event that varies between avians and non-avians is the fusion of bones, which happens earlier and more profusely." An earlier onset of character development during ontogeny (when compared to the ancestors) is a predisplacement, while a more profuse development can be the results of an acceleration and hypermorphosis. All three process are subcategories of peramorphosis. Thus, we are very confident that evolutionary changes in skull network configuration results from an extension (or acceleration) of ontogenetic change. That peramorphic processes affected the evolution of crown birds is also evident in the growth patterns of birds. If we compare the growth rates of modern birds with that of Mesozoic Enantiornithes, it is clear that modern birds have a speed up growth, which represents an acceleration. This example was only briefly mentioned in the previous version, but we extended it for the revision as further example of peramorphic evolution in birds.**

Also, on a more technical side: number of connections is expected to vary together with number of bones but a measure of that is not a measure of complexity (e.g., Esteve-Altava et al., 2014). Furthermore, density of connections could be interpreted as a measure of skull complexity and it increases significantly in adult birds as compared with juveniles and non-avians, therefore all references to a decrease in complexity in modern birds should be changed. This pattern coincides with previous work on the effect of Willinston's Law, in which previous authors identified that bone fusions and loses are coupled with an increase in density of connections, and they interpreted as a way of development to cope with the harmful effects of these re-organizations (Esteve-Altava et al., 2013).

- **This is simply a semantic issue. The reviewer is correct that density of connections is a measure for morphological complexity. Here, we actually refer the term complexity not to morphological complexity, but to modular complexity, in which a higher number of modules is more complex in terms of integration. To avoid confusion, we revised the relevant sections in the text.**

In fact, in my opinion, this paper should be restructured to stress that: 1) it advances our knowledge on the evolution of anatomical networks in birds over ontogeny and evolution, this is interesting and useful in its own right; also, 2) because it describes the same system that has been tied by previous convincing work to particular developmental process producing paedomorphosis, it represents a rare opportunity to test how those heterochronic changes affect connectivity patterns of involved structures; and finally, 3) the restructuring role of development in coping with bone losses and fusions (Esteve-Altava et al., 2013) and its connection with cranial kinesis and kinesis complexity in birds should be discussed more in depth. In fact, I think that the expected role of paedomorphic process in anatomical networks (bone fusions and losses), the balancing effect of development (augmenting connectivity between the remaining bones) and the role of kinesis (selecting those complex associations between groups of bones for functional roles) all point the same direction and could explain your results much better than the current unfounded claims of peramorphism.

- **We do not understand this, as the general structure of our manuscript already follows the suggestion of the reviewer: 1) We believe that we already pointed out that our study advances the knowledge of the evolution of anatomical networks in birds over ontogeny and evolution. Of course, we could stress this even more, but we think that a proper presentation and discussion of the results is more important than arm waving. 2) As discussed above, we do not agree that our results indicate a paedomorphic evolution. Furthermore, we do not see why the conclusion of a peramorphosis should prohibit any testing how heterochronic changes affect connectivity patterns of involved structures, especially as we already did it? 3) We are happy to extend the discussion on the evolution of cranial kinesis a little bit more.**

Other minor issues involve the discussion of results from variational modularity and integration. Although Felice & Goswami, 2017 reported a modular organization of variation in shape in the skull of birds in a follow up paper Felice et al., 2019 found that the skull of modern birds is more integrated than non-avian dinosaurs and crocodylians. There are also a number of studies in selected groups of birds that point towards an integrated skull (Bright et al., 2016, 2019; in diurnal raptors and parrots respectively). However, there are still important issues with the quantification of modularity and integration in geometric morphometrics (see e.g., Cardini, 2019) which are likely to be affecting variation in shape in the skull of birds and because there are not ideal solutions yet, these results should be taken cautiously.

- **In the previous version, we already added a short discussion about the variational modularity approach of Goswami in the supplementary information. However, we are happy to extend this discussion a little bit more, as the reviewer provided us with more useful literature.**

Lines 7-11: this phrase should be deleted or simplified 'Triggered by numerous fossil discoveries from the Late Jurassic and Early Cretaceous of China, the evolutionary transition from terrestrial non-avian theropods to volant birds has become a vital, interdisciplinary research field, often focusing on feather evolution and the origin of flight.' I think it detracts from the main message of the paper.

- **The abstract was completely rewritten to be in agreement with the journal format.**

Line 13: strong modular integration is ambiguous. You should state if you mean to variational or connectivity modularity/integration.

- **We are thankful for this statement.**

Lines 32, 33: change 'aberrant' for 'modified'

Line 57: 'including the stem-birds, Archaeopterys and Ichthyornis'

Line 59: The authors should explain the difference between Q-modules and S-modules, one statement should be fine in here as the rest is explained in the Methods section.

Lines 218-219: This phrase should be deleted as it is not its place in the Methods section: 'Ontogenetic changes in skull modularity of birds were investigated in a macroevolutionary context.'

Bibliography: Revise the spelling of 'Borja Esteve-Altava' and various other authors.

- **All suggestions were incorporated.**

Figures:

GENERAL: Although I do not consider this critical for publication of the paper I wanted to suggest a series of new visualizations that might improve the paper:

1) Biplots in the PCoA plots so we can see how particular variables account for how variation is distributed in the multidimensional space. Also, it will be great to have a PCo1 vs PCo3 plot and a PCo2 vs PCo3 plot, with overlain biplots.

- **We followed the suggestion of the reviewer and added biplots into the PCoA graph.**
- **According to our analyses, only the first two PCo's show an ontogenetic signal (see regression result). As we prefer showing only the relevant results in the main text, we see no point in showing plots with other PCo's. However, all PCoA results are given in the supplementary tables and are available for future analyses.**

2) A composite figure showing the relationship of, at least, the most important pairs of variables derived from the anatomical (for instance number of bones against density of connections). If the authors are concerned about not showing the role of size at the same time, they can plot it as symbol size very easily using functions within ggplot2, for instance. Plotting a phylogeny over these plots will be desirable, particularly as the authors have composite phylogenies with all the specimens including juveniles.

- **We do not see how these comparisons can help to back up our findings. We think this is rather a kind of distraction from our take-home message.**

Figure 1: To improve clarity I suggest adding silhouettes to the plots d, e, f coloured as the three groups they represent. Also, because DFA PCoA vectors' direction is arbitrary I suggest changing one or the others so the distribution of the groups in the plot is the same in the three.

- **We added the silhouettes as suggested by the reviewer. The axes of the DFA and PCoA were not changed, as they represent different types of analyses.**

Figure 2: I suggest drawing the palatine bones and portraying the ventral/dorsal aspects of a more reduced set of taxa. I understand that this might be difficult, an alternative solution to this might be placing the composite figure of pairs of network variables I suggested above instead.

- **Adding further views, would overload the figure. In addition, for most theropod dinosaurs no good palatal reconstructions are available.**

Bibliography:

Esteve-Altava, B., Marugán-Lobón, J., Botella, H. and Rasskin-Gutman, D., 2013. Structural constraints in the evolution of the tetrapod skull complexity: Williston's law revisited using network models. *Evolutionary Biology*, 40(2), pp.209-219.

Esteve-Altava, B., Marugán-Lobón, J., Botella, H. and Rasskin-Gutman, D., 2014. Random loss and selective fusion of bones originate morphological complexity trends in tetrapod skull networks. *Evolutionary Biology*, 41(1), pp.52-61.

Felice, R.N., Watanabe, A., Cuff, A.R., Noirault, E., Pol, D., Witmer, L.M., Norell, M.A., O'Connor, P.M. and Goswami, A., 2019. Evolutionary integration and modularity in the archosaur cranium. *Integrative and comparative biology*.

Hu, D., Young, N.M., Xu, Q., Jamniczky, H., Green, R.M., Mio, W., Marcucio, R.S. and Hallgrímsson, B., 2015. Signals from the brain induce variation in avian facial shape. *Developmental Dynamics*, 244(9), pp.1133-1143.

Smith-Paredes, D., Núñez-León, D., Soto-Acuña, S., O'Connor, J., Botelho, J.F. and Vargas, A.O., 2018. Dinosaur ossification centres in embryonic birds uncover developmental evolution of the skull. *Nature ecology & evolution*, 2(12), p.1966.

Reviewers' comments:

Reviewer #1 (Remarks to the Author):

The authors have thought about all my suggested. The ones they agreed with have been properly addressed. Others have been argued against and I will not insist on them. However, I would like to make remarks on how the authors decided to address my two (and most important) comments on the methodology, which I do not agree.

1) The authors decided to keep Q_{max} as a variable, regardless of my suggestion to use Parcellation instead (or even drop it). They tested various alternatives and found no differences in their results (which is fantastic), so they decided to keep Q_{max} just because it works fine.

I would like to note that my suggestion was not based on those grounds. Q_{max} can very well correlate as good as any parameter or fit the author's models just fine). I objected against using Q_{max} as a variable because mathematically it makes no sense to include it as a variable for comparison (for the reasons exposed in my first review). And that is independent of what others have done in the past (myself included), one should try to improve what others have done, not to stick with their mistakes, when we learn how to do things better. The authors miss an opportunity here to set the record straight and take due credit.

2) On my suggestion on compare their modularity outputs with those of a more sophisticated community detection algorithm, such as OSLOM, the authors reported that they have tried but did not manage to use the program suggested.

I grant that using this software requires some effort (e.g. install it in a Linux environment, but not much more). The software web-page includes detailed instructions of how to do it, and reference to key publications to use it and interpret its outputs; I also provided my own paper showing its application to skull networks. Note that reviewer #2 also points in this direction. However, the authors excuse themselves again by invoking what others have done or not done in the past, unwillingly to consider the benefits for their own study of making things better than in the past. Another missed opportunity in my opinion.

Although I disagree on these decisions, they do not damage in any way the great work done by the authors and the conclusions of the study.

Minor comments on the revised version:

Line 1. The first sentence of the abstract seems more a result outline than a statement for the general background for the study, as it mentions already results about the integration in terms of connectivity. Perhaps a more general statement on the biological impact of bone fusions instead?

Line 235. delete "the absence or". Connections (K) are only present connections. Another thing is the cells of the adjacency matrix where 0s/1s are used to code absence/presence of connections, respectively. Or instead, replace the whole sentence with "where rows and columns identify bones and matrix cells code the absence (0) or the presence (1) of a joint between two bones" (or something like this). Although that is already stated in line 239. Check for redundancies.

Line 260-261. "[...] which quantifies the number of connections between nodes within the same module". Add, "minus the expected number of connections if they were distributed at random among the same modules for the best of the possible partitions." This is a more correct textual definition of

Q_max.

This review is not anonymous.
Signed, Borja Esteve-Altava

Reviewer #2 (Remarks to the Author):

My main big concern is still the asymmetries:

line 191 "However, this left right asymmetry might have a true biological meaning in terms of module hierarchy, indicating that single modules are not fully independent from each other, but are still influenced by neighbouring modules."

This seems insufficient. One thing is that neighboring modules influence each other, another is that the modularity of the skull is completely asymmetric. The modularity of the skull includes elements that lie "in between modules", linking two sides of the skull and can not be effectively assigned to one or the other side. I expressed my concern about this issue before since these represented asymmetries can alter the ontogenetic/evolutionary trends you are investigating. For example, the juvenile and adult *Tarbosaurus* differ in the number of skull modules due IN PART to an asymmetry on the modulatory including the bones posterior of the temporal fenestra. If these asymmetries are not addressed and explained properly beyond what they mean in terms of the program limitations, is difficult to take them in terms of what they might mean in a compared ontogenetic/evolutionary frame.

line 150 "... the intensive bone fusion during bird ontogeny exaggerates the ancestral adult traits of non-avian theropods, indicating peramorphic heterochrony."

The comparison of an adult avian skull to a juvenile or any non-avian skull reveals a reduction in the number of modules, which makes sense considering the adult skull consists of extremely fused bones. However, an indication that the non-avian ontogenetic trend was to go from a high number towards a lower number of more integrated modules, relies on A) an insufficient number of ontogenetic samples of dinosaurs (only *Tarbosaurus* has a juvenile and adult) and B) skull modularity representations that are heavily asymmetric and could influence the number of modules.

(A) might require more sampling to solve, but species with specimens in different stages in ontogeny exist and should be included. I think that only if, without question you can demonstrate that the ancestral ontogeny for birds (that is, acquired at any given point during theropod evolutionary history) involved a reduction in number of modules and more integration, you can present the results as peramorphic.

I strongly advice to revise the asymmetric results and test whether some means of reducing the artifact effect, (considering only one side of the skull, vs one side + the other's side midline bones, etc.) have an effect in the ontogenetic and evolutionary trends you describe.

As a comment on the final conclusion...

line 207 "... our study shows that the genotype of crown birds still contains relics from their theropod ancestors, which are normally not expressed in the adult phenotype"

Rephrase this, not in terms of the genotype (of which this study does not have any data) but in terms of the phenotypic ontogenetic processes. These studies show that modern birds essentially still build themselves in the manner of their theropod ancestors but modifications in the timing and place of the building blocks result in the evolutionary novelty that is the avian skull.

Reviewer #3 (Remarks to the Author):

The ontogenetic and evolutionary patterns of connectivity of the skull shown by the authors are the product of a more profuse fusion of bones in crown birds than in the stem groups over ontogeny.

In order to ascertain if one pattern observed can be explained as the product of a particular process of heterochrony one needs to discard alternative explanations, in particular those that stem from previous well-established hypotheses.

In this case the pattern that needs to be explained is the more profuse bone fusion in birds. In my opinion, this pattern can be explained, rather than by invoking a novel hypothesis, with the current interpretation of progenesis, that is, the heterochronic process of truncation of ontogeny in birds compared with non-avians, that previous workers have proposed on the basis of changes in ontogenetic timespans and shape change in the skull across this transition (e.g., Bhullar et al., 2012).

Namely, if the ontogeny is truncated in birds, at the same ontogenetic time, bird bones are less mature and therefore more plastic than in the stem groups. This entails that they can more easily invade each other's ossification regions, generating bone fusions as the ontogeny goes by and the region becomes fully ossified. A nice example of this, can be observed in Smith-Paredes et al., 2018 in which the authors showed all the ossification centres of the periorbital bones of non-avians appear in early stage embryos of modern birds. As the ontogeny goes by, this ossification centres invade each other's region of proliferation resulting in the reduced number of periorbital adult bones that we see in crown birds. The sutures in modern birds are seen between the bones that never fuse, those that never collide substantially when they are immature, and at the end of their maturation form a suture. Therefore, I argue that the authors got the polarity of the characters over ontogeny wrong: the more mature character is the suture not the fusion. If you apply this change to their results then the conclusions change: adult crown birds have few sutures while immature crown birds have no sutures at all, whereas adults of the stem-groups have more sutures, and even the immature specimens of non-avians (which to be honest are very few in this sample) have more sutures than crown bird juveniles and arguably adult crown birds. This pattern is congruent with progenesis and results in a paedomorphic state in adult modern birds, being more similar to the juveniles of non-avians.

As simple as that: higher bone fusion in adult crown birds compared with outgroups likely results from the same progenesis that generates the changes in anatomical architecture and shortage of overall maturation time in birds. I think this difference in the pattern between my interpretation and the author's evidences that depending on which character is selected the pattern can be different and even opposite, and only by scrutinizing well the polarity of traits in ontogeny we can be sure about the pattern.

As I said before I think the article is an important contribution for the field, but the main interpretation is fundamentally wrong and I recommend reframing the paper.

Reviewers' responses:

Reviewer #1 (Remarks to the Author):

The authors have thought about all my suggested. The ones they agreed with have been properly addressed. Others have been argued against and I will not insist on them. However, I would like to make remarks on how the authors decided to address my two (and most important) comments on the methodology, which I do not agree.

1) The authors decided to keep Q_{max} as a variable, regardless of my suggestion to use Parcellation instead (or even drop it). They tested various alternatives and found no differences in their results (which is fantastic), so they decided to keep Q_{max} just because it works fine.

I would like to note that my suggestion was not based on those grounds. Q_{max} can very well correlate as good as any parameter or fit the author's models just fine). I objected against using Q_{max} as a variable because mathematically it makes no sense to include it as a variable for comparison (for the reasons exposed in my first review). And that is independent of what others have done in the past (myself included), one should try to improve what others have done, not to stick with their mistakes, when we learn how to do things better. The authors miss an opportunity here to set the record straight and take due credit.

2) On my suggestion on compare their modularity outputs with those of a more sophisticated community detection algorithm, such as OSLOM, the authors reported that they have tried but did not manage to use the program suggested.

I grant that using this software requires some effort (e.g. install it in a Linux environment, but not much more). The software web-page includes detailed instructions of how to do it, and reference to key publications to use it and interpret its outputs; I also provided my own paper showing its application to skull networks. Note that reviewer #2 also points in this direction. However, the authors excuse themselves again by invoking what others have done or not done in the past, unwillingly to consider the benefits for their own study of making things better than in the past. Another missed opportunity in my opinion.

Although I disagree on these decisions, they do not damage in any way the great work done by the authors and the conclusions of the study.

- **We are grateful that the reviewer accepts our responses regarding the parameter Q_{max} and OSLOM. Just two brief comments: (1) As stated in the previous response, we included the Parcellation as a new parameter and tested its evolutionary signal, which supported our previous findings. (2) Running the OSLOM software was not a real issue [one of the authors (CF) is actually a Linux user]. The problem was that we were not able to interpret the results properly and that asymmetries still remained in theropod dinosaurs. Not able to judge if this result is from a wrong application of the software or not, we decided to stick with the protocol of Werneburg. Why the asymmetries were retained is uncertain, but a proper investigation of this issue goes beyond the aim of the paper.**

Minor comments on the revised version:

Line 1. The first sentence of the abstract seems more a result outline than a statement for the general

background for the study, as it mentions already results about the integration in terms of connectivity. Perhaps a more general statement on the biological impact of bone fusions instead?

Line 235. delete "the absence or". Connections (K) are only present connections. Another thing is the cells of the adjacency matrix where 0s/1s are used to code absence/presence of connections, respectively. Or instead, replace the whole sentence with "where rows and columns identify bones and matrix cells code the absence (0) or the presence (1) of a joint between two bones" (or something like this). Although that is already stated in line 239. Check for redundancies.

Line 260-261. "[...] which quantifies the number of connections between nodes within the same module". Add, "minus the expected number of connections if they were distributed at random among the same modules for the best of the possible partitions." This is a more correct textual definition of Q_{max} .

- **All three suggested modifications were implemented.**

This review is not anonymous.
Signed, Borja Esteve-Altava

Reviewer #2 (Remarks to the Author):

My main big concern is still the asymmetries:

line 191 "However, this left right asymmetry might have a true biological meaning in terms of module hierarchy, indicating that single modules are not fully independent from each other, but are still influenced by neighbouring modules."

This seems insufficient. One thing is that neighboring modules influence each other, another is that the modularity of the skull is completely asymmetric. The modularity of the skull includes elements that lie "in between modules", linking two sides of the skull and cannot be effectively assigned to one or the other side. I expressed my concern about this issue before since these represented asymmetries can alter the ontogenetic/evolutionary trends you are investigating. For example, the juvenile and adult *Tarbosaurus* differ in the number of skull modules due IN PART to an asymmetry on the modularity including the bones posterior of the temporal fenestra. If these asymmetries are not addressed and explained properly beyond what they mean in terms of the program limitations, is difficult to take them in terms of what they might mean in a compared ontogenetic/evolutionary frame.

- **First, we would like to apologize that we only briefly responded to the criticism of the reviewer regarding module asymmetry in the previous round.**
- **We would like to start with a more general statement on how to deal with methodological limitations. Biologist and palaeontologist, for instance, both use phylogenetic analyses to reconstruct the relationship of organisms based on algorithms with dichotomous branching. From the perspective of population biology, we know that sister species are able to hybridize, creating new hybrid species. Such events, however, are not taken into consideration in cladistic algorithms. Should we stop using cladistics now and neglect all previous results of phylogenetic analyses, because the dichotomous model does not reflect the whole spectrum of speciation? No, instead, we believe that scientists have to accept that methods have limitations, and can only lead to simplified models of reality. The**

important thing is to document these limitations (as we did) so that methods can be improved in the future. However, this is beyond the scope of this study.

- We performed a literature search and note that a left-right asymmetry in skull module distribution is present in the network analyses of Werneburg et al. 2019 and Navarro-Díaz et al. 2019, but, unfortunately, these were neither reported nor discussed. As in our study, these asymmetries seem to be driven by the unpaired bones along the skull midline, which are often attributed to either one side of the skull (in other cases they form an unpaired braincase module).
- Module asymmetries are further present in two studies on skull musculature in primates/humans (Powell et al. 2018; Diago et al. 2019). Here, asymmetry is reported and discussed in the context of the evolution of facial expressions (which is well documented in studies on primate behaviour). Likewise, we could argue that the asymmetry found in theropod skulls is related to some biomechanical properties or developmental constraints. Unfortunately, studies on skull biomechanics and development in theropods are rather rare, making such statements purely speculative.
- The crucial question is, and the reviewer is right to ask, does the limitation of the method affects the outcome of our analysis? The short answer is, no. However, we want to point out that we only report the presence of module asymmetry, but do not make any ontogenetic/evolutionary statements regarding its presence. Instead, all ontogenetic/evolutionary interpretations are based on single parameters and the PCoA summary. As stated in the manuscript, all network parameters (except of parameter C) show the heterochronic signal we describe. It is retained, if we run a PCoA excluding the S and Q module (and Qmax) parameters (see Figure and Table below). Even if the PCoA includes only the two basic parameter N (number of bones) and K (number of bone contacts), the heterochronic signal is present. Therefore, we are very confident that our results are robust and that the signal is not an artefact of module asymmetry. Nevertheless, in the revised version, we discuss this issue in more detail, although we do not have a final answer to this problem.

PCo1	Slope	R ²	p-Value
PCoA (original) vs PCoA (-Q and S modules)	1.06	0.9946	<2e-16 ***
PCoA (original) vs PCoA (-Q and S modules, Qmax)	1.04	0.9928	<2e-16 ***
PCoA (original) vs PCoA (just N and K)	0.73	0.9461	<2e-16 ***
PCo2	Slope	R	p-Value
PCoA (original) vs PCoA (-Q and S modules)	0.814	0.8379	<2e-16 ***
PCoA (original) vs PCoA (-Q and S modules, Qmax)	0.801	0.8584	<2e-16 ***

- **The reviewer is correct in that the fusion of frontals during ontogeny leads to a reduction of modules in adult *Tarbosaurus*, mirroring the situation seen in crown birds. We included this example in the text.**

line 150 "... the intensive bone fusion during bird ontogeny exaggerates the ancestral adult traits of non-avian theropods, indicating peramorphic heterochrony."

The comparison of an adult avian skull to a juvenile or any non-avian skull reveals a reduction in the number of modules, which makes sense considering the adult skull consists of extremely fused bones. However, an indication that the non-avian ontogenetic trend was to go from a high number towards a lower number of more integrated modules, relies on A) an insufficient number of ontogenetic samples of dinosaurs (only *Tarbosaurus* has a juvenile and adult) and B) skull modularity representations that are heavily asymmetric and could influence the number of modules.

(A) might require more sampling to solve, but species with specimens in different stages in ontogeny exist and should be included. I think that only if, without question you can demonstrate that the ancestral ontogeny for birds (that is, acquired at any given point during theropod evolutionary history) involved a reduction in number of modules and more integration, you can present the results as peramorphic.

- **Being familiar with theropod systematics, phylogeny and morphology, one of the authors (CF) does not agree with this statement. Yes, it is correct that more juvenile material is available, but in order to include it into the sample for network analyses, the material has to be complete and more or less articulated/associated so that the bone contacts can be identified and scored. This is usually not the case. For instance, CF participated in the original description of the juvenile megalosauroid *Sciurumimus*, which represents one of the most complete known dinosaurs from Europe. However, despite its exceptional preservation, we did not add it to our sample, because the palatal morphology and parts of the braincase are obscured and thus cannot be scored (at least not until synchrotron data is available for this specimen). It was already hard to find a proper sample of adult non-avian theropods (which is less than 5% of all species known) that fulfill these criteria. We added a statement along these lines to the material and method section.**
- **We also agree with the reviewer that a greater number of ontogenetic series from theropod dinosaurs would help to support our hypothesis. However, we think that the interpretation of the presence of peramorphosis is justifiable for the following reason. As peramorphosis is present, if the adult descendent has a more developed trait than its adult ancestor, it is actually just sufficient to know the adult condition of the ancestor. Detailed knowledge of the ancestral ontogeny is only necessary to classify subtypes of peramorphosis (hypermorphosis, acceleration or predevelopment), e.g., if the peramorphosis is caused by longer period of growth, a higher growth rate or an earlier onset of development. This is of course different for the second type of heterochrony, the paedomorphosis, where knowledge of the ancestral ontogeny is crucial to detect potential ancestral juvenile traits in the adult descendent. In sum, our sample is appropriate to detect peramorphic evolution itself, but not its subtypes, while it has limited explanatory power for the detection of paedomorphosis due to the small sample of juveniles (see also reviewer#3).**

I strongly advice to revise the asymmetric results and test whether some means of reducing the artifact effect, (considering only one side of the skull, vs one side + the other's side midline bones, etc.) have an effect in the ontogenetic and evolutionary trends you describe.

- **We did a case analysis for *Archaeopteryx*, which shows a left-right asymmetry in the distribution of the modules. The original result was compared with three alternative scenarios: a) only one side of the skull without unpaired midline bones, b) one side of the skull plus unpaired midline bones and c) both sides of the skulls without unpaired midline bones. Analysing only one side of the skull results in a distribution that resembles the modularity of the left side of the *Archaeopteryx* in the original analysis, but with jugal and quadratojugal forming an own zygomatic module. When the midline bones of the midline are included in the network analysis, the quadrate is also part of this zygomatic module. However, when both skull sides without unpaired midline bones are analysed the skull modules become symmetric, resembling the pattern of the left side in the original analysis. This latter case supports our argument that the asymmetry is driven by unpaired bones that imbalance the topology of the cluster dendrogram. As previously stated, exploring the**

impact of module asymmetry is beyond the scope of the paper, but if the reviewer is interested, we invite him/her to contact us after the current manuscript is accepted, and help us to design a case study where we go deeper into the subject.

As a comment on the final conclusion...

line 207 "... our study shows that the genotype of crown birds still contains relics from their theropod ancestors, which are normally not expressed in the adult phenotype"

Rephrase this, not in terms of the genotype (of which this study does not have any data) but in terms of the phenotypic ontogenetic processes. These studies show that modern birds essentially still build themselves in the manner of their theropod ancestors but modifications in the timing and place of the building blocks result in the evolutionary novelty that is the avian skull.

- **We are thankful for the suggestion. The sentence was modified.**

Reviewer #3 (Remarks to the Author):

The ontogenetic and evolutionary patterns of connectivity of the skull shown by the authors are the product of a more profuse fusion of bones in crown birds than in the stem groups over ontogeny.

- **We agree!**

In order to ascertain if one pattern observed can be explained as the product of a particular process of heterochrony one needs to discard alternative explanations, in particular those that stem from previous well-established hypotheses. In this case the pattern that needs to be explained is the more profuse bone fusion in birds. In my opinion, this pattern can be explained, rather than by invoking a novel hypothesis, with the current interpretation of progenesis, that is, the heterochronic process of truncation of ontogeny in birds compared with non-avians, that previous workers have proposed on the basis of changes in ontogenetic timespans and shape change in the skull across this transition (e.g., Bhullar et al., 2012).

- **As stated in the manuscript and in the previous review response, we do not deny that the skull evolution of birds (in particular its shape) was impacted by paedomorphic heterochrony. Indeed, one of the authors (CF), has previously demonstrated that the paedomorphic shape evolution detected by Bhullar roots actually deeper in the theropod evolution (see Foth et al. 2016). What we hypothesize in the current study is that within this paedomorphic shape evolution, there is an oppositional peramorphic heterochrony that affects the bone configuration.**
- **The example given by the reviewer is a nice opportunity to demonstrate the dual presence of opposite heterochronies in the evolution of birds. But before we start: Not the ontogeny of birds is truncated, but its period of growth/maturation (ontogeny describes the span from fertilization till death and some birds can become pretty old. In general, birds become older than similarly sized mammals). Nevertheless, the reviewer is missing a little detail in his/her argument. When we compare the growth rates of crown group birds with that of Mesozoic birds and non-avian theropods, modern birds grow significantly faster than their ancestors do. For example, the basal Ornithomorph *Archaeohynchus*, which has the size of a quail, needed more than 3 years to reach somatic maturity (Wang et al. 2017 J.**

Syst. Palaeont. 15:1-18), while the common quail *Coturnix coturnix* achieves this in less than two months (Blotzheim et al. 1994. Handbuch der Vögel Mitteleuropas, Bd. 5, Galliformes and Gruiformes). By definition, rate increase is an acceleration, which is a subcategory of peramorphosis (Klingenberg 1998, Biol. Rev. 73: 79-123). Thus, it is correct that the truncated time of somatic maturation in extant birds is most likely a paedomorphosis when compared with their stem, but the speed-up of maturation itself represents a peramorphosis. We added this example to the discussion to support our hypothesis. We would like to highlight that findings of similar oppositional heterochronies abound in the literature for other groups of vertebrates. We humans, for instance, are universally acknowledged to have peramorphic brain growth, but pedomorphic facial growth relative to our ancestors. Us proposing oppositional heterochronies for birds is therefore uncontroversial in itself.

- Finally, the study of Wang et al. 2017, which investigates the evolution of fusion in the avialan manus and pelvis, concluded that fusion events have a high plasticity and are not correlated with the growth patterns of basal birds. Consequently, we believe that the assumption of the reviewer represents rather a (untested) hypothesis than a fact.

Namely, if the ontogeny is truncated in birds, at the same ontogenetic time, bird bones are less mature and therefore more plastic than in the stem groups. This entails that they can more easily invade each other's ossification regions, generating bone fusions as the ontogeny goes by and the region becomes fully ossified.

- If this argument were correct, we would observe more bone fusion in small theropods than in large ones, as the former have shorter maturation times and immature bones that can potentially fuse. In reality, we see the opposite. Frequent bone fusion is rather found in large-bodied theropods (e.g., tyrannosaurids, carcharodontosaurids and abelisaurids). Crown birds represent an exception.
- As shown by our study, the majority of bone fusion happens after hatching and not during embryogenesis, i.e., when the ossification centres have already become bones. At the time of hatching, the bones are not fully developed, indeed, but this is logical, because birds (like other amniotes) still have to grow. However, at this stage bones need to be functional, allowing movements and feeding. This is especially true for birds with precocial lifestyle, which is the plesiomorphic condition in crown birds (Starck & Ricklefs 1998).

A nice example of this, can be observed in Smith-Paredes et al., 2018 in which the authors showed all the ossification centres of the periorbital bones of non-avians appear in early stage embryos of modern birds. As the ontogeny goes by, this ossification centres invade each other's region of proliferation resulting in the reduced number of periorbital adult bones that we see in crown birds.

The sutures in modern birds are seen between the bones that never fuse, those that never collide substantially when they are immature, and at the end of their maturation form a suture. Therefore, I argue that the authors got the polarity of the characters over ontogeny wrong: the more mature character is the suture not the fusion. If you apply this change to their results then the conclusions change: adult crown birds have few sutures while immature crown birds have no sutures at all, whereas adults of the stem-groups have more sutures, and even the immature specimens of non-avians (which to be honest are very few in this sample) have more sutures than crown bird juveniles and arguably adult crown birds. This pattern is congruent with progenesis and results in a paedomorphic state in adult modern birds, being more similar to the juveniles of non-avians.

- The study of Smith-Paredes et al. is a very important contribution to the understanding the evolution of bird skulls, but we think that the reviewer is overinterpreting their results. The study of Smith-Paredes et al. shows that extant birds still have temporary ossification centres for the postorbital and prefrontal (bones that are present in non-avian theropods, but absent in the crown birds) during the embryonic development. During embryogenesis, these centres fuse with their neighbouring ossifications centres so that postorbital and prefrontal do not develop as independent structures. Instead, they are integrated into the frontal and nasal bones, respectively. Before fusion, however, the relevant centres need obtain physical contact with each other. As they are not fully developed bones, one can argue, if this temporary contact zone should be called a suture or not, but this is purely a semantic issue. The important aspect is that no fusion can happen, if the centres do not meet! It is also important to realize that the study document only two (or three) fusion events between bone precursors (taking the additional coronoid fusion into account) that happen during embryogenesis. As stated above, the majority of fusion events (those we documented in our study) happens after hatching when ossification centres have become real bones. In the figure below, one can see the skull of a juvenile (left) and adult (right) Eagle owl (*Bubo bubo*) in occipital view. As it is pretty obvious, the single bones of the juvenile bird can be identified very easily due to sutures. In the adult bird, however, they cannot be identified anymore, because all sutures have disappeared due to fusion. Thus, we absolutely do not agree with the character polarity suggested by reviewer: The more developed character state is the fusion (see also Bailleul et al. 2016). The polarity is also supported by observations made from theropod evolution, where fusion of bones always represents the apomorphic character state. We have modified the discussion.

As simple as that: higher bone fusion in adult crown birds compared with outgroups likely results from the same progenesis that generates the changes in anatomical architecture and shortage of overall maturation time in birds. I think this difference in the pattern between my interpretation and the author's evidences that depending on which character is selected the pattern can be different and even opposite, and only by scrutinizing well the polarity of traits in ontogeny we can be sure about the pattern.

- We agree that character polarity is the key to understanding ontogenetic and evolutionary processes. However, based on our arguments presented above, we are very confident that in terms of character polarity, bone fusion represents the most advanced stage in both the ontogeny and evolution, as it requires a previous bone contact via sutures. Thus, we are also confident that our interpretation of the heterochronic patterns is correct.

As I said before I think the article is an important contribution for the field, but the main interpretation is fundamentally wrong and I recommend reframing the paper.

- **If the reviewer believes that our interpretation is fundamentally wrong, we invite him to write a response to our manuscript after publication and let the scientific community decide.**

REVIEWERS' COMMENTS:

Reviewer #3 (Remarks to the Author):

Although I am overall not convinced by the arguments presented by the authors, and we do not know much about rates of ontogenetic growth in stem-birds (as osteohistology has proven to be very variable, even within the same specimen, and difficult to interpret), I am willing to accept the argument the authors made about oppositional heterochronies. It is true that even if our ontogenetic data for stem birds, including non-avian dinosaurs, is not precise, one can argue confidently that growth is accelerated in some way modern birds. If this is the case then, a process generating accelerated ontogenesis could produce peramorphic traits. However, processes producing paedomorphism and peramorphism could not be synchronic in the evolution of a lineage, but sequential. Therefore, I think the paper will benefit from stating very clearly, from the abstract, that peramorphism happened very close to the crown origin while paedomorphism affected avian and avialans more broadly.

Response to reviewers:

Reviewer #3 (Remarks to the Author):

Although I am overall not convinced by the arguments presented by the authors, and we do not know much about rates of ontogenetic growth in stem-birds (as osteohistology has proven to be very variable, even within the same specimen, and difficult to interpret), I am willing to accept the argument the authors made about oppositional heterochronies. It is true that even if our ontogenetic data for stem birds, including non-avian dinosaurs, is not precise, one can argue confidently that growth is accelerated in some way modern birds. If this is the case then, a process generating accelerated ontogenesis could produce peramorphic traits. However, processes producing paedomorphism and peramorphism could not be synchronic in the evolution of a lineage, but sequential. Therefore, I think the paper will benefit from stating very clearly, from the abstract, that peramorphism happened very close to the crown origin while paedomorphism affected avian and avialans more broadly.

- **Many thanks for the debate. We are happy that we could partly convince the reviewer. The suggested statement were implemented in the abstract, introduction (final paragraph) and discussion, emphasising that both heterochronies happened not simultaneously but successive.**